# FEWER BATTLES, MORE GAIN: AN INFORMATION-EFFICIENT FRAMEWORK FOR ARENA-BASED LLM EVALUATION

**Zirui Liu**[1], **Xianquan Wang**[1], **Yan Zhuang**[3], **Jiatong Li**[1], **Qi Liu**[1,2*],
**Shuanghong Shen**[2], **Mingyue Cheng**[1], **Shijin Wang**[4]

1: State Key Laboratory of Cognitive Intelligence, University of Science and Technology of China
2: Institute of Artificial Intelligence, Hefei Comprehensive National Science Center
3: College of Artificial Intelligence, Nanjing University of Aeronautics and Astronautics
4: iFLYTEK AI Research (Central China), iFLYTEK Co., Ltd
`{liuzirui,wxqcn,zykb,cslijt}@mail.ustc.edu.cn`
`{qiliuql,mycheng}@ustc.edu.cn`
`shshen@iai.ustc.edu.cn, sjwang3@iflytek.com`

## ABSTRACT

Arena-based evaluation has become a key method for assessing large language models (LLMs) through head-to-head model comparisons, closely reflecting human preferences. However, current arena rating systems (e.g., ELO rating system) often suffer from inefficiencies due to exhaustive or random model pair annotations, leading to redundant evaluations, longer evaluation times, and lower overall efficiency. To address these challenges, we propose a novel adaptive model-pair selection algorithm. By leveraging the asymptotic normality of LLM ability estimation under sparse conditions, our approach strategically selects high-value model pairs, focusing on confrontations with the lowest variance. Specifically, we introduce Fisher information as a metric to guide model pair selection, optimizing the evaluation process through A-optimality and D-optimality. A-optimality minimizes estimation variance, ensuring balanced reliability across models, while D-optimality reduces uncertainty by maximizing the determinant of the Fisher Information Matrix. Extensive experiments on both simulated and real-world datasets demonstrate that our method outperforms existing approaches in terms of information efficiency and result reliability. Notably, our method offers a flexible, general toolkit that can be easily integrated into existing arena-based platforms, greatly improving scalability and efficiency for large-scale LLM evaluations. Our code is publicly available to promote reproducibility at `https://github.com/Liuz-rui/Adaptive-Arena`.

## 1 INTRODUCTION

The rapid advancements in large language models (LLMs) have led to their widespread use in various fields Li et al. (2025); Jiang et al. (2025), including content generation (Wang et al., 2024a), customer service, and intelligent tutoring (Kasneci et al., 2023). As LLM architectures (Liu et al., 2024a), fine-tuned variants (Hu et al., 2023; Cheng et al., 2025b), and domain-specific models (Patil & Gudivada, 2024; Cheng et al., 2026; Pan et al., 2026; Cheng et al., 2025a) continue to evolve, thus the demand for efficient and scalable evaluation frameworks is growing (Mizrahi et al., 2024; Zhou et al., 2023; Li et al., 2024b; Zhuang et al., 2025a). Among existing evaluation methods, arena-based evaluation is notable (Chiang et al., 2024): unlike traditional benchmark-based approaches (Wang et al., 2024b; Mirzadeh et al., 2024) that rely on predefined prompts and fixed answers, arena-based frameworks assess models through head-to-head confrontations. This approach captures real-world performance dynamics and produces rankings that closely align with human judgments (Zheng et al., 2023; Luo et al., 2024), making it a valuable tool for assessing LLMs' practical utility.

---

*Corresponding Author.

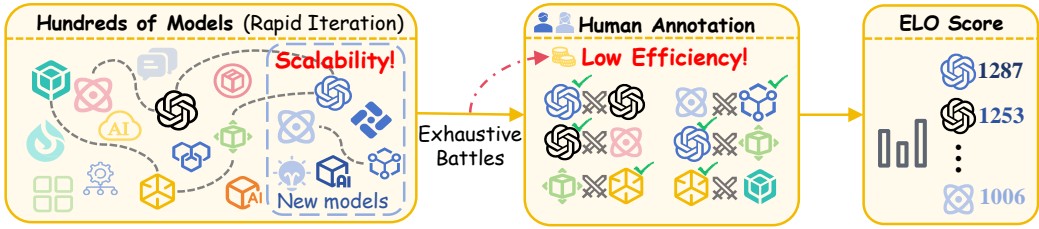

Figure 1: The diagram of Arena-based Evaluation. Models engage in pairwise battles, annotators label the outcomes, and the ELO ratings are then estimated based on these annotations.

However, current arena-based evaluation methods face significant issues. First, is the **efficiency** (Luo et al., 2024; Spangher et al., 2025). As shown in Figure 1, these frameworks often rely on exhaustive or random pairwise annotations to generate reliable rankings, which becomes increasingly resource-intensive. Second, is the **scalability**. As the number of newly proposed models grows, platforms like Chatbot Arena (Chiang et al., 2024), hosting over 190 models, require tens of thousands of confrontations to produce reliable rankings (Patil & Gudivada, 2024). This not only consumes considerable resources but also results in substantial evaluation cycles, making it difficult to keep up with the rapid development of LLMs (Min et al., 2025).

To address these challenges, we propose a novel adaptive arena-based evaluation method. Leveraging the asymptotic normality of ability estimation under sparse conditions, we can strategically select high-value model pairs for evaluation and improve efficiency by minimizing the variance of the asymptotic distribution. Specifically, we introduce two key methods: A-optimality, which minimizes the variance in model estimates, and D-optimality, which reduces uncertainty by maximizing the determinant of the Fisher Information Matrix (Rissanen, 1996; Ly et al., 2017). Through these techniques, we achieve *more gain* in each annotation, leading to *fewer battles*.

Our contributions are as follows:

- We first introduce the concept of efficiency in arena-based LLM evaluation, using statistical uncertainty to minimize redundant evaluations, significantly improving both speed and resource usage in ranking.

- We apply A-optimality and D-optimality to model-pair selection, backed by rigorous theoretical analysis, to reduce annotation costs while preserving ranking reliability.

- Through extensive experiments, we show that our method outperforms existing approaches in efficiency and reliability, and can seamlessly integrate with current arena-based platforms. This makes our method not just a theoretical innovation, but a practical and scalable solution for large-scale LLM evaluations.

## 2  RELATED WORK

**Arena-based Evaluation**  Arena-based evaluation is a key approach in LLM evaluation, distinguished from traditional benchmark methods by its focus on direct head-to-head model competitions (Wang et al., 2025). This design provides a more realistic measure of models' relative abilities (Zellers et al., 2019; Hendrycks et al., 2020; Cobbe et al., 2021). Unlike traditional approaches that use predefined datasets (Liang et al., 2022; Liu et al., 2024b), arena settings emphasize dynamic model confrontations that better reflect real-world usage scenarios. Ranking systems are central to Arena-based evaluation (Busa-Fekete et al., 2014; Szörényi et al., 2015; Chernoff, 1992), with most methods relying on the ELO Rating System to estimate model proficiency (Coulom, 2007; Pelánek, 2016). However, ELO's sensitivity to sample order can lead to instability in performance estimates (Boubdir et al., 2024). To address this, recent studies have proposed adaptations, such as statistical refinements (Ameli et al., 2024; Yin et al., 2024; Liu et al., 2025) and bias-mitigation methods like UDA (Zhang et al., 2025). Notably, all existing Arena-based methods aim to improve ranking accuracy but overlook the critical issue of annotation efficiency. Whether using standard ELO, adapted variants, or frameworks like UDA, these approaches often involve unnecessary confrontations, leading to low annotation efficiency and high evaluation overhead.

**Efficiency of LLM Evaluation** Efficiency has become a key focus in LLM evaluation due to the high costs of annotating large datasets and conducting extensive tests (Marion et al., 2023). Existing methods (Xie et al., 2023; Chung et al., 2023; Saranathan et al., 2024; Li et al., 2024a; Zhuang et al., 2025b) share the goal of reducing the number of evaluation examples while maintaining accuracy, but they are designed primarily for traditional static question-answering (QA) tasks, where models are assessed against predefined prompts and answers. For example, Polo et al. (2024a) introduced TinyBenchmarks, which evaluates LLMs with fewer examples than traditional benchmarks by prioritizing information-dense samples, thus reducing annotation and computational costs while maintaining reliable performance estimation. Similarly, Kipnis et al. (2024) proposed MetaBench, which selects diverse, compact prompts based on statistical analysis, avoiding redundant evaluations while capturing core model abilities. Based on this, Polo et al. (2024b) developed an efficient multi-prompt evaluation method, which optimizes prompt selection across rounds to reduce redundant data collection and computation without sacrificing accuracy. However, these efficiency-driven methods are not applicable to dynamic Arena-based evaluation for two key reasons: First, Arena scenarios involve pairwise head-to-head model confrontations rather than static QA tasks; second, rankings in Arena settings rely on iterative human judgments (Kruglanski & Ajzen, 1983), breaking the assumption of fixed criteria inherent in traditional QA methods. Therefore, the dynamic and interactive nature of Arena evaluation makes current efficiency methods unsuitable.

Given these gaps—existing efficiency methods being incompatible with dynamic Arena scenarios, and Arena-based methods neglecting annotation efficiency—our study aims to address these issues. We propose optimizing Arena-based evaluation through strategic sample selection techniques to both accelerate system convergence and improve overall evaluation efficiency.

## 3 ADAPTIVE ARENA METHOD

### 3.1 MODELING ELO RATINGS

Arena-based evaluation has emerged as a key approach for LLM evaluation, where models compete head-to-head on benchmarks or datasets, with outcomes annotated by evaluators. Let $\mathcal{S} = \{(i, j, w_{ij}) | i < j \in [N]\}$ represent the comparative records, where $N$ is the total number of models being evaluated. Each element $(i, j, w_{ij}) \in \mathcal{S}$ contains important information: it indicates that model $i$ and model $j$ have engaged in a competition, and $w_{ij}$ is the annotation result. Specifically, if $w_{ij} = 1$, model $i$ wins; if $w_{ij} = 0$, model $j$ wins; and if $w_{ij} = 0.5$, it indicates a tie. The goal of arena-based evaluation is to estimate the ranking scores $(r_1, \ldots, r_N)$. Since the ranking is relative, we assume that the ability of the last model is 0, *i.e.*, $r_N = 0$, and in the following, we focus on the ability estimation of the first $N - 1$ models, denoted as $\boldsymbol{r} = (r_1, \ldots, r_{N-1})$.

To model the win probabilities, we use the Bradley-Terry model (Rao & Kupper, 1967), which forms the statistical foundation for modern ELO-style rating systems. It posits that the probability of model $i$ winning against model $j$, denoted $P_{ij}$, is a logistic function of their ability difference:

$$P_{ij} = \frac{1}{1 + e^{-C(r_i - r_j)}}, \tag{1}$$

where $C$ is a scaling constant.

With the obtained winning probability, many methods can be adopted to get the ranking scores. Among these, Maximum Likelihood Estimation (MLE)-based **ELO** finds the ratings that best explain all observed match results simultaneously. It is a more statistically robust batch approach to estimate the abilities $\boldsymbol{r}$ by MLE. It is achieved by minimizing the negative log-likelihood function:

$$\hat{\boldsymbol{r}} = \arg\min_{\boldsymbol{r}} \mathcal{L}_{\mathcal{S}}(\boldsymbol{r}), \quad \text{where} \quad \mathcal{L}_{\mathcal{S}}(\boldsymbol{r}) = -\sum_{(i,j,w_{ij}) \in \mathcal{S}} [w_{ij} \ln P_{ij} + w_{ji} \ln P_{ji}]. \tag{2}$$

Adopting the MLE framework is crucial as it allows us to rigorously analyze the statistical properties of the estimated ratings, which is essential for developing our adaptive selection strategy. The detailed properties are as follows.

**Statistical Properties of ELO.** Consider modeling the arena as an Erdős-Rényi (E-R) Graph $G(N, q_N)$ (Kumar et al., 2000; Guimera et al., 2004), where $N$ is variable and the sample size $|\mathcal{S}|$

increases with $N$. Here, $q_N$ represents the probability that a confrontation occurs between models $i$ and $j$. Under this framework, the estimation results of ELO exhibit both uniqueness and asymptotic normality (Simons & Yao, 1999; Han et al., 2020):

**Lemma 1.** *Let $G(N, q_N)$ be the Erdős-Rényi Graph, and $\hat{r}_n, r_n^*$ be the estimated and true abilities of the first $n$ models, respectively. We have:*

*(1) (**Uniqueness**) If $q_N = \omega\left(\frac{\log N}{N}\right)$, then ELO almost surely has a unique solution $\hat{r}_{N-1}$.*

*(2) (**Asymptotic Normality**) If $q_N \cdot \frac{N^{1/10}}{(\log N)^{1/5}} \to \infty$ as $N \to \infty$, then for any $n < N$, $\sqrt{|\mathcal{S}|}(\hat{r}_n - r_n^*) \xrightarrow{d} \mathcal{N}(0, \mathcal{I}_{\mathcal{S}}(r_n^*)^{-1})$, where $\mathcal{I}_{\mathcal{S}}(r^*)$ is the **Fisher information matrix**.*

In the ELO method, Fisher information (Frieden, 2004) is calculated by solving the Hessian matrix:

$$\mathcal{I}_{\mathcal{S}}(r) = -\mathbb{E}[\mathcal{H}(\mathcal{L}_{\mathcal{S}}(r))] = C^2 \sum_{(i,j,w_{ij}) \in \mathcal{S}} P_{ij} P_{ji} (e_i - e_j)(e_i - e_j)^\top, \qquad (3)$$

where $e_i$ is the $i$-th standard basis vector, and $e_N = 0$. To explain the Fisher information matrix intuitively, consider a toy example with three models, $X$, $Y$, and $Z$, and two battle records $\mathcal{S} = \{(X > Y), (X = Z)\}$. Assuming all models have an initial ability of 0 and that model $Z$ is fixed during the estimation process, the calculation of the information quantity is as follows:

$$C^2 \begin{bmatrix} 0 & 0 \\ 0 & 0 \end{bmatrix} \xrightarrow{X > Y} C^2 \begin{bmatrix} 0.25 & -0.25 \\ -0.25 & 0.25 \end{bmatrix} \xrightarrow{X = Z} C^2 \begin{bmatrix} 0.5 & -0.25 \\ -0.25 & 0.25 \end{bmatrix}, \qquad (4)$$

and we also provide a case study of the information matrix based on real dataset in Appendix E.

Considering Lemma 1 with Fisher information matrix like Equation 4, it can be confirmed that ELO has a unique estimated solution and this estimation is asymptotically unbiased and normally distributed, even in sparse scenarios. Therefore, from a statistical perspective, ***the efficiency problem in arena evaluation essentially revolves around how to reduce variance more quickly*** during the evaluation process. Specifically, the key lies in **reducing the inverse of the Fisher information matrix $\mathcal{I}_{\mathcal{S}}(r^*)$** under ELO context.

### 3.2 Selection Strategy for Arena-based Evaluation

As mentioned earlier, our direct goal is to reduce the inverse of the information matrix. However, the true ability values of candidate models, $r^*$, are unknown, making it impossible to directly select the model pairs. Therefore, we dynamically select each subsequent model pair based on the accumulated records $\mathcal{S}_{t-1}$. Specifically, by approximating the true abilities $r^*$ with the estimated abilities $\hat{r}^{t-1}$ derived from $\mathcal{S}_{t-1}$, we can compute an approximate information matrix. Using the Fisher information of the current abilities, we propose the following A-optimality method (Chan, 1982; Wanyonyi et al., 2021) for Arena-based Evaluation:

**The A-optimality method.** This method aims to select the model pair that minimizes the sum of the sampling variances of the ability estimators, which is equivalent to minimizing the trace of the inverse of the information matrix:

$$(i_t, j_t) = \arg \min_{1 \leq i < j \leq N} \text{tr}\left[\left(\mathcal{I}_{\mathcal{S}_{t-1}}(\hat{r}^{t-1}) + \mathcal{I}_{\{(i,j)\}}(\hat{r}^{t-1})\right)^{-1}\right]. \qquad (5)$$

For this method to be feasible, the Fisher information matrix must be invertible. We have the following theorem:

**Theorem 1.** *Consider the Erdős-Rényi Graph $G(N, q_N)$ for ELO. If $q_N = \omega\left(\frac{\log N}{N}\right)$, then the information matrix $\mathcal{I}_{\mathcal{S}}(r)$ is almost surely positive definite.*

From Theorem 1, it follows that as long as there are sufficient battle samples, the information matrix is guaranteed to be positive definite (Johnson, 1970). This not only ensures that the information matrix is invertible but also guarantees that the determinant is positive.

However, A-optimality method still requires calculating the inverse of a matrix, which increases computational complexity. To address this, we consider maximizing the determinant of information matrix and propose the D-optimality method (John & Draper, 1975) for Arena-based Evaluation:

Table 1: This table summarizes the characteristics of each strategy. Using $\hat{r}$ indicates whether the strategy utilizes currently estimated abilities; Global Info indicates whether global information is used, and $\min q_N$ represents the minimum edge probability that ensures effective selection.

| Method | Random | Nearest | Ada-Pair | A-optimality | D-optimality |
|---|---|---|---|---|---|
| Using $\hat{r}$ | ✗ | ✓ | ✓ | ✓ | ✓ |
| Global Info | ✗ | ✗ | ✗ | ✓ | ✓ |
| $\min q_N$ | 0 | $w(\frac{\log N}{N})$ | 1 | $w(\frac{\log N}{N})$ | $w(\frac{\log N}{N})$ |
| Complexity | $O(1)$ | $O(N)$ | $O(N^2)$ | $O(N^{\beta+3})$ | $O(N^{\beta+2})$ |

**The D-optimality method.** The D-optimality method aims to select the model pair that maximizes the determinant of the information matrix in the current state, expressed as:

$$(i_t, j_t) = \arg \max_{1 \le i < j \le N} \left| \mathcal{I}_{\mathcal{S}_{t-1}}(\hat{r}^{t-1}) + \mathcal{I}_{\{(i,j)\}}(\hat{r}^{t-1}) \right|. \tag{6}$$

### 3.3 Theoretical Analysis

In this subsection, we analyze our proposed method from three perspectives: algorithmic prerequisites, numerical computation, and time complexity.

**Algorithmic Prerequisites.** Different selection strategies depend on distinct prerequisites. Table 1 summarizes the conditions for each method.

For example, the Random selection method requires no additional variables, so it is often used to initialize other strategies. In contrast, methods that consider model abilities $\hat{r}$ must meet at least the conditions outlined in Lemma 1(1), which represents the theoretical lower bound of $q_N$ for these methods. More stringent conditions apply to methods like Ada-Pair, which considers the confidence interval of each model pair $(i, j)$. To ensure the existence of confidence intervals, Ada-Pair requires that each pair has competed at least once, implying that the density of the E-R graph must satisfy $q_N = 1$. This requirement becomes restrictive as the number of models $N$ increases. This highlights a key advantage of our proposed method: it only requires the minimal condition, $\omega(\frac{\log N}{N})$, to ensure the positive definiteness of the information matrix (Theorem 1) without requiring additional samples for initialization. This feature demonstrates the broad applicability of our approach.

**Numerical Computation.** To avoid numerical instability and ensure computational validity, we need to consider the relationship between Fisher information and the number of samples. We present the following theorem to address this:

**Theorem 2.** *Let $r$ be the ability parameter vector, and let $\mathcal{I}_{\mathcal{S}_t}(r)$ denote the Fisher information matrix based on the subset $\mathcal{S}_t$ with size $t$. Define:*

- $\mathcal{A}(\mathcal{S}_t) = \frac{1}{tr[\mathcal{I}_{\mathcal{S}_t}(r)^{-1}]}$ *(average information efficiency),*
- $\mathcal{D}(\mathcal{S}_t) = |\mathcal{I}_{\mathcal{S}_t}(r)|^{\frac{1}{N-1}}$ *(normalized geometric information density).*

*Assume that each element in $\mathcal{S}_t$ contributes bounded positive information and that $\mathcal{I}_{\mathcal{S}_t}(r)$ is positive definite for sufficiently large $t$. Then as $t \to \infty$:*

1. *$\mathcal{A}(\mathcal{S}_t) = O(t)$ (there exist constants $C_1$ and $t_1$ such that $\mathcal{A}(\mathcal{S}_t) \le C_1 \cdot t$ for all $t \ge t_1$),*
2. *$\mathcal{D}(\mathcal{S}_t) = O(t)$ (there exist constants $C_2$ and $t_2$ such that $\mathcal{D}(\mathcal{S}_t) \le C_2 \cdot t$ for all $t \ge t_2$).*

Theorem 2 provides the upper bound for the variation of Fisher information with respect to the number of competitions. Specifically, the inverse of the A-optimality measure grows linearly with the increase in the number of samples, while the $(N-1)$-th root of the D-optimality measure grows linearly. Therefore, when practically implementing the D-optimality method, it is advisable to transform the calculation into the form defined in Theorem 2 to prevent numerical explosion.

**Time Complexity.** Algorithm 1 presents the pseudo-code for model selection in the Arena. The main computational cost in the model pair selection phase comes from calculating the determinant or

---

**Algorithm 1:** The Adaptive Arena Framework

---

**Initialize:** Step $t \leftarrow 1$, Model's ability $\hat{\boldsymbol{r}}^0$, Battle records $\mathcal{S}_0 \leftarrow \emptyset$.

**1 while** True **do**

**2**      Select model pair: $(i_t, j_t) \leftarrow \arg\max_{1 \leq i < j \leq N} \left| \mathcal{I}_{\mathcal{S}_{t-1}}(\hat{\boldsymbol{r}}^{t-1}) + \mathcal{I}_{\{(i,j)\}}(\hat{\boldsymbol{r}}^{t-1}) \right|$ .

**3**      Get Annotation for model pair: $w_{i_t j_t} \leftarrow Anno(i_t, j_t)$.

**4**      Add record in $\mathcal{S}$: $\mathcal{S}_t \leftarrow \mathcal{S}_{t-1} \cup \{(i_t, j_t, w_{i_t j_t})\}$.

**5**      Update model's ability $\hat{\boldsymbol{r}}$: $\hat{\boldsymbol{r}}^t \leftarrow \arg\max_{\boldsymbol{r}} \mathcal{L}_{\mathcal{S}_t}(\boldsymbol{r})$, $t \leftarrow t + 1$.

     **Output:** The models' estimated abilities $\hat{\boldsymbol{r}}^t$.

---

the inverse of the matrix. Both operations have a complexity of $O(N^\beta)$ (Kaltofen & Villard, 2005), where $N$ is the matrix size or the number of models in the arena. Therefore, for the D-optimality method, which traverses all model pairs and computes the determinant of the information matrix at each step, the complexity is $O(N^{\beta+2})$. For the A-optimality method, whether calculating the trace of the inverse matrix or computing $N$ submatrices, it requires $N$ additional calculations per iteration, resulting in a complexity of $O(N \times N^2 \times N^\beta) = O(N^{\beta+3})$. The complexity for calculating the information matrix involves summing over $t$ iterations, making the overall time complexity for the A-optimality method $O(|S|^2 + |S| \cdot N^{\beta+3})$, and for the D-optimality method, $O(|S|^2 + |S| \cdot N^{\beta+2})$.

Although our method has a higher time complexity than previous approaches, its primary goal is to reduce human-annotated samples by leveraging computational time, ultimately minimizing the overall annotation effort. Therefore, we only require our method to meet the operational needs of the Arena system—providing model pairs for evaluation within a limited time for each user request. To achieve this, we have implemented the following improvements: (1) The information matrix is updated directly by adding the current model pair's information to the matrix from the previous step. (2) To address system concurrency, the top-$K$ model pairs with the highest information gain are selected for each request.

We discuss the time cost and the efficiency of these improvements in section 4.2. Although our method does not directly reduce the complexity related to the annotation volume, its value becomes evident within the iterative, ongoing Arena framework. For example, if our method improves efficiency by 50%, it means that the annotation results from our approach can achieve the same outcomes as twice the original annotation volume, given the Arena's continuous operation. This offers a significant long-term advantage, enabling comparable or even superior results with fewer resources over extended periods.

## 4 EXPERIMENTS

In this section, we evaluate the performance of the proposed A-optimality and D-optimality methods through experiments on both simulated and real-world datasets, analyzing the abilities of each method and discussing the time cost for proposed method.

### 4.1 EXPERIMENTAL SETUP

**Evaluation Metric.** Since the ELO score is relative, we use the pairwise index (Hastie & Tibshirani, 1997) to measure the consistency between the estimated abilities and the true abilities:

$$Pairwise = \frac{2}{N(N-1)} \sum_{1 \leq i < j \leq N} \mathbb{I}\left[(r_i^* - r_j^*)(\hat{r}_i - \hat{r}_j) > 0\right], \tag{7}$$

where $\mathbb{I}[\cdot]$ is the indicator function, $\boldsymbol{r}^*$ is true abilities of the models, and $\hat{\boldsymbol{r}}$ is the estimated abilities. Since the true abilities are not available in real-world scenarios, we use *the results estimated from the full dataset by ELO* (Equation 2) as the ground truth (Zhuang et al., 2023; Liu et al., 2024c). Lemma 1 ensures that $\boldsymbol{r}^*$ derived from this process is unique and unbiased, making this ground truth feasible. In other words, our task can be viewed as selecting a subset of matchups that achieves the same evaluation quality as using the full set, thereby significantly improving efficiency.

Table 2: Performance of Pairwise Metrics for Different Strategies. Results are averaged over five experiments and four steps. Bold values indicate significant improvements over the baseline.

| Method | | Dataset | | Is Simu | | ELO Method | | Total |
|--------|---------|---------|--------|--------|--------|--------|--------|--------|
| | | Chatbot | PPE | Real | Simu | ELO | m-ELO | |
| Original | Random | 0.8797 | 0.8021 | 0.8501 | 0.8316 | 0.8379 | 0.8439 | 0.8409 |
| | Nearest | 0.8825 | 0.7995 | 0.8422 | 0.8398 | 0.8412 | 0.8407 | 0.8410 |
| | Ada-Pair | 0.8835 | 0.8040 | 0.8452 | 0.8423 | 0.8379 | 0.8496 | 0.8437 |
| | A-optim | 0.8880 | 0.8033 | 0.8494 | 0.8418 | 0.8406 | 0.8506 | 0.8456 |
| | D-optim | **0.8941** | **0.8122** | **0.8596** | **0.8466** | **0.8486** | **0.8577** | **0.8531** |
| Improved | Nearest | 0.8897 | 0.8020 | 0.8525 | 0.8392 | 0.8351 | 0.8566 | 0.8459 |
| | Ada-Pair | 0.8849 | 0.7993 | 0.8509 | 0.8332 | 0.8376 | 0.8465 | 0.8421 |
| | A-optim | 0.8807 | 0.8061 | 0.8488 | 0.8380 | 0.8372 | 0.8495 | 0.8434 |
| | D-optim | **0.8918** | **0.8243** | **0.8581** | **0.8581** | **0.8546** | **0.8616** | **0.8581** |

**Dataset.** We conducted experiments on two real-world datasets: **Chatbot** (Chiang et al., 2024; Zheng et al., 2023) and **PPE** (Frick et al., 2024). The Chatbot dataset[1] was collected from 13,000 distinct IP addresses on the Chatbot Arena platform, while the PPE dataset[2] contains human preference evaluations for Preference Proxy Evaluations. Detailed statistics and experimental settings are provided in Appendix C.1.

**Compared Approaches.** This paper compares two model ability evaluation methods: **ELO** (Albers & Vries, 2001) and **m-ELO** (Liu et al., 2025). ELO refers to the traditional ELO, it updates each model's rating iteratively by adding the difference between the annotation $w_{ij}$ and the prediction $P_{ij}$, while m-ELO uses a MLE process like Equation 2. For model pair selection, we compare several common strategies used in both traditional competitions and modern LLM arenas. These strategies include: **Random**, which randomly selects a model pair for annotation at each step and serves as a baseline; **Nearest**, which selects the two models with the closest abilities at each step, a typical strategy in competitive games; and **Ada-Pair** (Chiang et al., 2024), an official selection strategy from Chatbot Arena, which takes into account the confidence interval of each model pair and selects the one that maximizes the reduction of the interval. Additionally, for each method, we consider both the original version and the time-improved version discussed in Section 3.3, referred to as **Original** and **Improved**, respectively.

## 4.2 EXPERIMENT RESULTS AND ANALYSIS

To evaluate the effectiveness of our proposed selection method, we conducted experiments on real-world datasets. The following conclusions were drawn:

**Performance on Pairwise Metrics.** Figure 2(a) shows the performance of different selection strategies at each step, and Table 2 presents the average pairwise for each strategy. Our proposed method, D-optim, outperforms all others across all scenarios and is the only method that significantly surpasses the Random baseline. Additionally, as shown in Figure 2(a), D-optim effectively reduces variance in most cases, indicating that it lowers evaluation uncertainty and enhances efficiency.

The Nearest method, on the other hand, performs suboptimally in the LLM arena. This is due to the nature of the arena: rather than actual battles, models answer a single question, making ties more likely than in traditional competitions. This diminishes the effectiveness of pairing models with similar abilities. We also observe that while m-ELO incurs additional time costs, its impact on selection strategies that rely on current model abilities is notable. In practical applications, engineers need to balance computational efficiency and accuracy, such as using m-ELO for high-precision tasks and ELO in resource-constrained settings.

**Performance of Time-Improved Method on Pairwise Metrics.** To assess the impact of our improvements on proposed methods, we compared the Nearest, Ada-Pair, A-optimality, and D-

---

[1] https://huggingface.co/datasets/lmsys/chatbot_arena_conversations
[2] https://huggingface.co/datasets/lmarena-ai/PPE-Human-Preference-V1

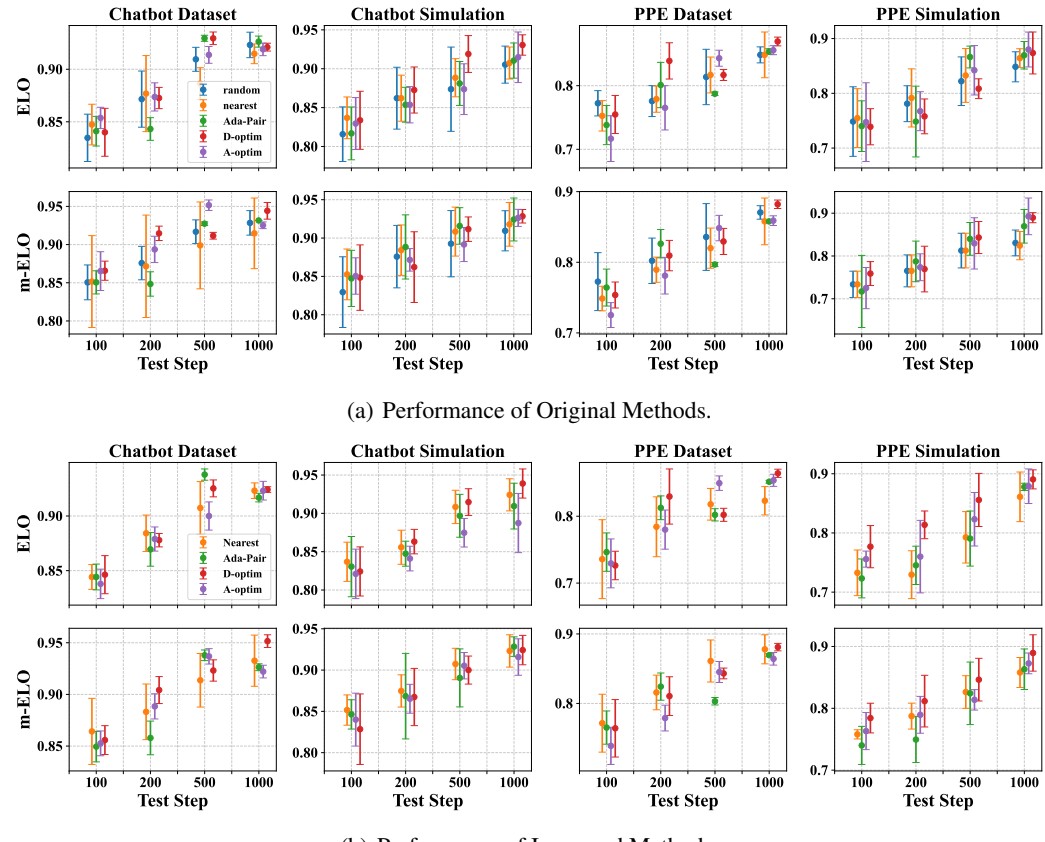

(a) Performance of Original Methods.

(b) Performance of Improved Methods.

Figure 2: This Error bar illustrates pairwise metrics for each dataset and ELO method. "Dataset" indicates randomly sampled competition results, while "Simulation" denotes results simulated based on real model abilities. Error bars represent standard deviation.

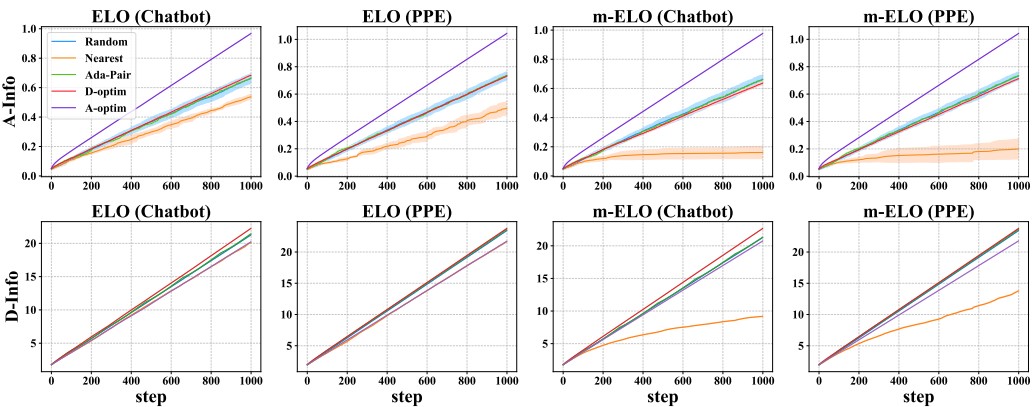

Figure 3: This line chart shows A-Info and D-Info for each dataset and ELO method.

optimality methods, where the Top-10 samples were selected in each sampling round (i.e., $K = 10$). The experimental results are presented in Figure 2(b) and Table 2.

We can observe that our D-optimality method not only maintains its original performance but also shows improvements, where a total 0.5% gain is observed compared to the time-improved version. This suggests that model pairs with higher D-Info are indeed prioritized for selection, supporting the effectiveness of our D-Info design. This result confirms the effectiveness of our improved D-optim. In the appendix C.3, we also present its performance on larger-scale datasets.

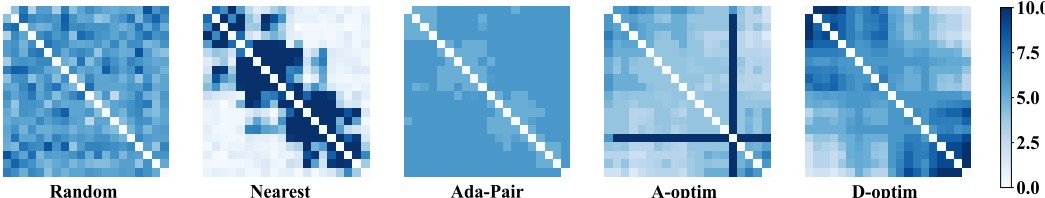

Figure 4: Heatmap of model pair sampling frequency for different selection methods. Each cell represents the number of battles between Model $i$ (ranked by ability) and Model $j$ (ranked by ability).

**Performance on Fisher Information Gain.**  Figure 3 shows the variation of $\mathcal{A}(\mathcal{S}_t) = \frac{1}{\mathrm{tr}[\mathcal{I}_{\mathcal{S}_t}(\boldsymbol{r})^{-1}]}$ (A-Info) and $\mathcal{D}(\mathcal{S}_t) = |\mathcal{I}_{\mathcal{S}_t}(\boldsymbol{r})|^{\frac{1}{N-1}}$ (D-Info) as the number of samples increases. As shown in Theorem 2, both A-Info and D-Info exhibit linear growth. The experimental results confirm that under most strategies, their growth is indeed linear. Furthermore, our method achieves the fastest improvement rate (steepest slope) in both A-Info and D-Info. In contrast, Ada-Pair shows similar information gain to the Random method. This suggests that our method, despite using estimated abilities, **enhances information gain** from samples, improving evaluation efficiency.

**Visualization of Sampling Distribution.**  To visualize the sampling distribution of model pairs selected by each strategy, we sorted the models by true abilities and used heatmaps to represent the sampling frequency of each pair. Figure 4 shows that D-optim selects model pairs with similar true abilities more often (darker colors near the diagonal), while ensuring diverse coverage. In contrast, Ada-Pair samples more uniformly across all pairs, and Nearest focuses too much on similar models.

Interestingly, the A-optimality method results in many samplings involving a specific model, since this model's ability is fixed in the ELO system. This creates a unique role for the model pair selection: The A-optimality method, while ensuring sparsity (i.e., most pairs have battled), prefer to use this model as an "anchor" to estimate others' abilities. As shown in Figure 3, this method actually leads to high A-Info, but may introduce bias and impact the Arena's sustainability. An improvement could be to fix each model sequentially, calculate A-Info for each, and average the results, but this would increase complexity. In contrast, D-optimality is minimally impacted by this constraint by ELO, making D-Info a more suitable metric for Arena-based evaluation.

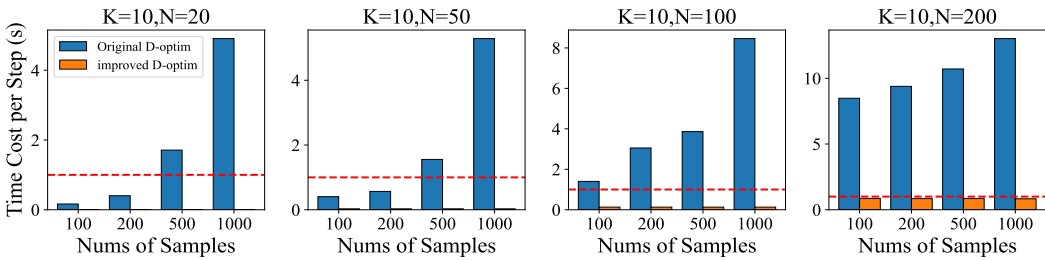

Figure 5: Time costs for Original D-optimality and Improved D-optimality Methods as the number of samples and models increases.

**Performance on Time Cost.**  Figure 5 reports the time required for each sample selection as the number of samples grows, with the red line marking 1 second. Although the D-optimality method does not always achieve the absolute minimum time, its cost remains far lower than that of LLM API calls or human annotations in realistic settings. Given the relatively low computational cost of our method, in practice, we can leverage computation to reduce the reliance on human annotation wherever possible. These results suggest that our method is efficient in practice and can be seamlessly incorporated into existing arena-based evaluation platforms.

## 5  CONCLUSION

In this work, we addressed the efficiency bottleneck in arena-based LLM evaluation. By leveraging the asymptotic normality of ELO-based ability estimation and introducing Fisher information as a guiding principle, we proposed adaptive model-pair selection strategies grounded in A-optimality

and D-optimality. These methods systematically reduce redundant battles while preserving ranking reliability, thereby accelerating evaluation and lowering annotation costs. Extensive experiments on both simulated and real-world datasets confirm that our framework consistently outperforms existing approaches in efficiency and robustness. Beyond improving scalability for current arenas, our approach can be regarded as a general and flexible toolkit that can be readily integrated into future evaluation platforms. We believe this work opens a promising direction toward information-efficient, scalable, and reliable assessment of large language models. In future work, an important direction is to incorporate instance-level factors into the m-ELO framework, paving the way for fully adaptive evaluation across test instances, models, and even annotators.

## 6 ACKNOWLEDGMENTS

This work was supported by grants from the National Key Research and Development Program of China (Grant No. 2024YFC3308200), the National Natural Science Foundation of China (U25B2072, 62502486, 62507010), the Key Technologies R&D Program of Anhui Province (No. 202423k09020039), the Fundamental Research Funds for the Central Universities and the Anhui Provincial Higher Education Digital Transformation Project.

## 7 ETHICS STATEMENT

This research does not involve any human subjects, animals, or sensitive data. The proposed methods and experiments focus solely on the evaluation of LLMs using publicly available datasets. We ensure that all data used in our experiments are publicly accessible, adhering to ethical standards and relevant data usage policies. Furthermore, our work does not raise any concerns related to bias, fairness, or social impact, as it is focused on improving the efficiency of model evaluations in a controlled, academic setting. In summary, our research does not present any ethical issues, and we are committed to conducting future work in line with ethical best practices and standards.

## 8 REPRODUCIBILITY STATEMENT

To ensure the reproducibility of our results, we provide the complete implementation of our methods, along with all necessary code and data, on GitHub repository. The repository includes scripts to replicate the experiments presented in the paper, including data preprocessing, model training, and evaluation procedures. All hyperparameters, random seeds, and environment configurations are specified to ensure consistent results across different platforms. The repository can be accessed at the following link:`https://github.com/Liuz-rui/Adaptive-Arena`.

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

## A Proofs of Theorem 1

**Theorem 1.** *Consider the Erdős-Rényi Graph $G(N, q_N)$, if $q_N = \omega(\frac{logN}{N})$, then the infomation matrix $\mathcal{I}_{\mathcal{S}}(\boldsymbol{r})$ is almost surely positive definite.*

*Proof.* Given that $q_N = \omega(\frac{logN}{N})$ implies that the graph $G(N, q_N)$ is almost surely connected (Guimera et al., 2004; Chatterjee & Varadhan, 2011), we next prove the following: if there exists a path between any two models $i$ and $j$, then the information matrix is positive definite. Assume $r_N = 0$ and consider the remaining variables $\boldsymbol{r} = (r_1, \cdots, r_{N-1})$. Consider the log-likelihood function for records $\mathcal{S}$:

$$\mathcal{I}_{\mathcal{S}}(\boldsymbol{r}) = C^2 \sum_{(i,j,w_{ij}) \in \mathcal{S}} P_{ij} P_{ji} (\boldsymbol{e}_i - \boldsymbol{e}_j)(\boldsymbol{e}_i - \boldsymbol{e}_j)^\top \tag{8}$$

where $\boldsymbol{e}_i$ is the $i$-st standard basis vector and let $\boldsymbol{e}_N = \boldsymbol{0}$. Let the number of matches between model $i$ and model $j$ be $\delta_{ij}$ and define $a_{ij} = \delta_{ij} C^2 P_{ij} P_{ji}$, then the quadratic form (O'Meara, 2013) for $\mathcal{I}_{\mathcal{S}}(\boldsymbol{r})$ can be expressed as:

$$\boldsymbol{x}\mathcal{I}_{\mathcal{S}}(\boldsymbol{r})\boldsymbol{x}^\top = \sum_{1 \le i < j \le N-1} a_{ij}(x_i - x_j)^2 + \sum_{i=1}^{N-1} a_{iN} x_i^2 + \sum_{j=1}^{N-1} a_{jN} x_j^2.$$

Note that $a_{ij} \ge 0$, therefore:

$$\boldsymbol{x}\mathcal{I}_{\mathcal{S}}(\boldsymbol{r})\boldsymbol{x}^\top \ge 0.$$

The following proves that the condition for the equality of this inequality can only be achieved at $\boldsymbol{0}$.

Suppose there exists a non-zero vector $\hat{\boldsymbol{x}} = (\hat{x}_1, \cdots, \hat{x}_{N-1})$ such that $\hat{\boldsymbol{x}}\mathcal{I}_{\mathcal{S}}(\boldsymbol{r})\hat{\boldsymbol{x}}^\top = 0$. Without loss of generality, assume that $\hat{x}_1 \ne 0$. Then, since the graph formed by the pairwise battles among the models is connected, it means that there must exist a path $(c_0 = 1, c_1, c_2, \cdots, c_{k-1}, c_k = N)$ that connects 1 and $N$. Then $a_{c_i c_{i+1}} > 0$, from which we can deduce that $\hat{x}_1 = \hat{x}_{c_1} = \cdots = \hat{x}_{c_{k-1}}$. Since $c_{k-1}$ and $N$ are connected, it indicated that $a_{c_{k-1}N} > 0$. Therefore, $\hat{x}_{c_{k-1}} = 0$, and thus $\hat{x}_1 = 0$, which is a contradiction! So $\boldsymbol{x}\mathcal{I}_{\mathcal{S}}(\boldsymbol{r})\boldsymbol{x}^\top = 0$ if and only if $\boldsymbol{x}$ is the zero vector. Therefore, $\mathcal{I}_{\mathcal{S}}(\boldsymbol{r})$ is a positive definite matrix (Johnson, 1970). $\square$

## B Proofs of Theorem 2

**Lemma 2.** *Let $0 < \lambda_1^t \le \lambda_2^t \le \cdots \le \lambda_N^t$ denote the eigenvalues of the information matrix $\mathcal{I}_{\mathcal{S}_t}(\boldsymbol{r})$, then $\lambda_N^t = O(t)$.*

*Proof.* Without loss of generality, we can assume that model $i$ and $j$ compete against each other at time $t + 1$. It is not hard to see that the information increment $C^2 P_{ij} P_{ji}(e_i - e_j)(e_i - e_j)^\top$ is a rank-1 matrix and its only positive eigenvalue $a$ satisfies that $0 < a \le 2C^2 P_{ij} P_{ji} \le C^2$. Therefore, by Weyl's Inequality for matrix perturbations (Weyl, 1949; Franklin, 2000), the eigenvalues of the information matrix at time $t+1$ satisfy $\lambda_N^t \le \lambda_N^{t+1} \le \lambda_N^t + a \le \lambda_N^t + C^2$. By recursively expanding this inequality, we obtain that $\lambda_N^t \le \lambda_N^{t-1} + C^2 \le \cdots \le N \cdot C^2$, i.e. $\lambda_N^t = O(t)$. $\square$

**Theorem 2.** *Let $\mathcal{A}(\mathcal{S}_t) = \frac{1}{tr[\mathcal{I}_{\mathcal{S}_t}(\boldsymbol{r})^{-1}]}$ and $\mathcal{D}(\mathcal{S}_t) = |\mathcal{I}_{\mathcal{S}_t}(\boldsymbol{r})|^{\frac{1}{N-1}}$, then it can be proved that: (1)$\mathcal{A}(\mathcal{S}_t) = O(t)$ (2)$\mathcal{D}(\mathcal{S}_t) = O(t)$.*

*Proof.* Since $tr(\mathcal{I}_{\mathcal{S}_t}(\boldsymbol{r})^{-1}) = \frac{1}{\lambda_1^t} + \frac{1}{\lambda_2^t} + \cdots + \frac{1}{\lambda_N^t} \ge \frac{N}{\lambda_N^t}$ and $|\mathcal{I}_{\mathcal{S}_t}(\boldsymbol{r})| = \lambda_1^t \lambda_2^t \cdots \lambda_N^t \le (\lambda_N^t)^N$, we can derive the following bounds for the optimality criteria:

For the A-optimality criterion $\mathcal{A}(\mathcal{S}_t) = \frac{1}{tr[\mathcal{I}_{\mathcal{S}_t}(\boldsymbol{r})^{-1}]} \le \frac{\lambda_N^t}{N}$.

For the D-optimality criterion $\mathcal{D}(\mathcal{S}_t) = |\mathcal{I}_{\mathcal{S}_t}(\boldsymbol{r})|^{\frac{1}{N-1}} \le \lambda_N^t$.

By Lemma 2, we know $\lambda_N^t = O(t)$, substituting this into the above inequalities yields $\mathcal{A}(\mathcal{S}_t) = O(t)$ and $\mathcal{D}(\mathcal{S}_t) = O(t)$. $\square$

## C  DETAILS OF EXPERIMENTS

### C.1  STATISTICS OF THE DATASET

In addition to the two datasets (**Chatbot** and **PPE**) mentioned in the main text, we have also included a larger dataset **Arena-100k** (Chiang et al., 2024; Tang et al., 2025) in the appendix. The Arena-100k[1] dataset comprises conversation data collected from June 2024 to August 2024, as well as English human preference evaluations, which were used to develop Arena Explorer. Each record in datasets includes a question ID, the names of the two competing models, their full conversation transcripts and the annotator's vote. The statistics of the datasets can be seen in Table 3.

Table 3: Statistics of the dataset

| Dataset | **Chatbot** | **PPE** | **Arena-100k** |
|---|---|---|---|
| #Models | 20 | 20 | 55 |
| #Response logs | 33000 | 16038 | 106134 |
| #Response logs per model | 1650.0 | 801.9 | 1929.7 |
| #Response logs per model pair | 173.68 | 89.10 | 71.47 |

### C.2  DETAILS OF EXPERIMENTAL SETTING.

In this experiment, 100 sample pairs will be randomly selected during the model initialization phase to initialize ELO, and the performance of different selection strategies will be recorded at steps 100, 200, 500, and 1000. After a selection strategy chooses a model pair, we consider two methods to simulate the competition result: (1) Conducting without-replacement random sampling on the real competition results in the dataset; (2) Simulating based on the real model abilities. If the abilities of the two models are $r_i^*, r_j^*$ and $p = \sigma(C(r_i^* - r_j^*))$, then each battle will return 1 with a probability of $p^2$, 0 with a probability of $(1-p)^2$ and 0.5 with a probability of $2p(1-p)$. As for ELO rating system, we set the hyperparameter $C$ of ELO to $\frac{\log 10}{400}$ and epochs to 50 for each battle in m-ELO. All experiments described in this paper can be executed on one 8G NVIDIA GeForce RTX 4060.

### C.3  PERFORMANCE ON LARGE DATASET

To further validate the effectiveness of the D-optimality method, we also conducted experiments on a larger dataset, namely Arena-100k. The key difference from the previous experiments lies in the following: as the number of models increased to 55, the number of initial samples we used was also increased to 1000. All other experimental settings remained unchanged. The experimental results are presented in Figure 6 and Table 4.

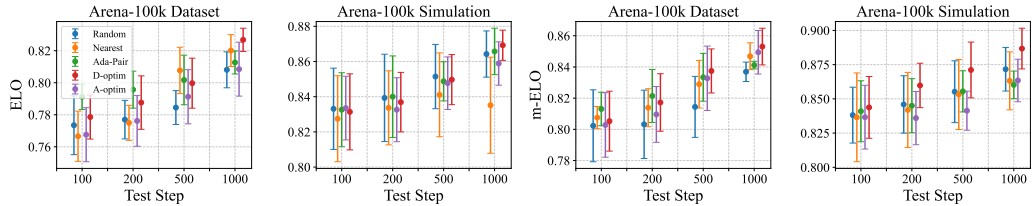

Figure 6: This line chart illustrates the pairwise metrics for Arena-100k dataset and each ELO method. Here, "Dataset" indicates that the competition results are randomly sampled from the dataset, while "Simulation" denotes that the results are simulated based on the real model abilities. The error bars represent the standard deviation.

It can be observed that with an increase in the number of models, the errors introduced by the traditional ELO method into the information matrix become amplified, leading to inaccurate information estimation. This explains why the D-optim method exhibits superior performance specifically in terms of m-ELO. However, at step 1000, the D-optim method achieves the best performance across all experimental settings. Therefore, overall speaking, as long as the current abilities of the models can be effectively estimated, the D-optimality method remains fully effective even in scenarios with a relatively large number of models.

---

[1]https://huggingface.co/datasets/lmarena-ai/arena-human-preference-100k

Table 4: The Performance of pairwise for Different Selection Algorithms. Each experimental result represents the average of five experiments and four selection steps, the bold font represents a significant improvement in statistics compared to the baseline.

| Method | Is Simu | | ELO Method | | Total |
|---|---|---|---|---|---|
| | Real | Simu | ELO | m-ELO | |
| Random | 0.8000 | 0.8499 | 0.8164 | 0.8335 | 0.8249 |
| Nearest | 0.8083 | 0.8415 | 0.8133 | 0.8365 | 0.8249 |
| Ada-Pair | **0.8138** | 0.8486 | **0.8236** | 0.8388 | 0.8312 |
| A-optim | 0.8047 | 0.8438 | 0.8145 | 0.8340 | 0.8243 |
| D-optim | 0.8132 | **0.8561** | 0.8225 | **0.8468** | **0.8346** |

# D  USAGE OF LLM

According to the rules about LLM use set by the ICLR 2026 conference, when we worked on this paper, we only used LLMs to polish the text. This includes making the flow of language smoother and improving how well sentences connect logically. We did not use LLMs for key steps like coming up with main points, making research content, or handling data. Also, we made sure that the polishing process does not change the main ideas or weaken the academic strictness of the original text, and we fully follow the standards for proper LLM use set by ICLR 2026.

# E  CASE STUDY OF FISHER INFOMATION MATRIX

To facilitate a better understanding of our method, we directly visualize the changes in the information matrix throughout the experimental process. Specifically, we adopt the D-optimality method: in each iteration, a model pair is selected, and the model ability estimates are updated in real time. Below, we present the information matrix of 5 iterations under the condition of 100 initial samples:

$$
\begin{pmatrix}
3.7 & -0.4 & -0.2 & -0.2 & 0.0 & 0.0 & 0.0 & 0.0 & -0.5 & -0.2 & -0.2 & -0.5 & 0.0 & -0.5 & -0.2 & -0.2 & 0.0 & 0.0 & -0.2 \\
-0.4 & 2.3 & 0.0 & 0.0 & 0.0 & 0.0 & 0.0 & 0.0 & 0.0 & -0.5 & 0.0 & -0.4 & -0.2 & 0.0 & -0.2 & 0.0 & 0.0 & 0.0 & -0.2 \\
-0.2 & 0.0 & 2.0 & -0.2 & 0.0 & 0.0 & 0.0 & -0.5 & 0.0 & 0.0 & -0.2 & 0.0 & 0.0 & 0.0 & -0.2 & -0.2 & 0.0 & 0.0 & -0.2 \\
-0.2 & 0.0 & -0.2 & 2.9 & 0.0 & -0.2 & 0.0 & -0.2 & -0.2 & -0.2 & -0.4 & 0.0 & 0.0 & -0.5 & -0.2 & 0.0 & 0.0 & -0.2 & 0.0 \\
0.0 & 0.0 & 0.0 & 0.0 & 1.0 & 0.0 & 0.0 & 0.0 & 0.0 & 0.0 & 0.0 & -0.2 & 0.0 & 0.0 & -0.2 & -0.2 & -0.2 & 0.0 & 0.0 \\
0.0 & 0.0 & 0.0 & -0.2 & 0.0 & 1.6 & -0.2 & 0.0 & 0.0 & 0.0 & 0.0 & 0.0 & 0.0 & 0.0 & -0.2 & -0.4 & 0.0 & 0.0 & -0.2 \\
0.0 & 0.0 & -0.5 & 0.0 & 0.0 & -0.2 & 2.7 & -0.5 & 0.0 & 0.0 & -0.4 & -0.2 & -0.4 & 0.0 & -0.2 & 0.0 & 0.0 & 0.0 & 0.0 \\
0.0 & 0.0 & 0.0 & -0.2 & 0.0 & 0.0 & -0.5 & 2.7 & -0.2 & -0.2 & 0.0 & 0.0 & 0.0 & 0.0 & 0.0 & -0.4 & -0.4 & 0.0 & -0.2 \\
-0.5 & 0.0 & 0.0 & -0.2 & 0.0 & 0.0 & 0.0 & -0.2 & 1.6 & 0.0 & 0.0 & 0.0 & -0.4 & 0.0 & 0.0 & 0.0 & 0.0 & 0.0 & 0.0 \\
-0.2 & -0.5 & -0.2 & -0.2 & 0.0 & 0.0 & 0.0 & -0.2 & 0.0 & 2.7 & -0.2 & -0.6 & 0.0 & 0.0 & 0.0 & 0.0 & 0.0 & -0.2 & 0.0 \\
-0.2 & 0.0 & 0.0 & -0.4 & 0.0 & 0.0 & -0.4 & 0.0 & 0.0 & -0.2 & 2.8 & -0.4 & 0.0 & -0.2 & -0.7 & 0.0 & 0.0 & 0.0 & 0.0 \\
-0.5 & -0.4 & 0.0 & 0.0 & -0.2 & 0.0 & -0.2 & 0.0 & 0.0 & -0.6 & -0.4 & 3.1 & -0.2 & 0.0 & 0.0 & -0.2 & -0.2 & 0.0 & 0.0 \\
0.0 & -0.2 & 0.0 & 0.0 & 0.0 & 0.0 & -0.4 & 0.0 & -0.4 & 0.0 & 0.0 & -0.2 & 1.9 & -0.2 & -0.5 & 0.0 & 0.0 & 0.0 & 0.0 \\
-0.5 & 0.0 & -0.2 & -0.5 & 0.0 & 0.0 & 0.0 & 0.0 & 0.0 & 0.0 & -0.2 & 0.0 & -0.2 & 1.6 & 0.0 & 0.0 & 0.0 & 0.0 & 0.0 \\
-0.2 & -0.2 & -0.2 & -0.2 & 0.0 & -0.2 & -0.2 & -0.2 & 0.0 & 0.0 & 0.0 & -0.7 & 0.0 & -0.5 & 3.1 & 0.0 & -0.2 & 0.0 & 0.0 \\
-0.2 & 0.0 & 0.0 & 0.0 & -0.2 & -0.4 & 0.0 & -0.4 & 0.0 & 0.0 & 0.0 & -0.2 & 0.0 & 0.0 & 0.0 & 2.2 & -0.2 & -0.2 & 0.0 \\
0.0 & 0.0 & 0.0 & 0.0 & -0.2 & 0.0 & 0.0 & -0.4 & 0.0 & 0.0 & 0.0 & -0.2 & 0.0 & 0.0 & -0.2 & -0.2 & 1.8 & 0.0 & -0.2 \\
0.0 & 0.0 & 0.0 & -0.2 & 0.0 & 0.0 & 0.0 & 0.0 & 0.0 & -0.2 & 0.0 & 0.0 & 0.0 & 0.0 & 0.0 & -0.2 & 0.0 & 0.9 & -0.2 \\
-0.2 & -0.2 & -0.2 & 0.0 & 0.0 & -0.2 & 0.0 & -0.2 & 0.0 & 0.0 & 0.0 & 0.0 & 0.0 & 0.0 & 0.0 & 0.0 & -0.2 & -0.2 & 2.1
\end{pmatrix}
$$

↓  Step 1:(17,4,1)

$$
\begin{pmatrix}
3.7 & -0.4 & -0.2 & -0.2 & 0.0 & 0.0 & 0.0 & 0.0 & -0.5 & -0.2 & -0.2 & -0.5 & 0.0 & -0.5 & -0.2 & -0.2 & 0.0 & 0.0 & -0.2 \\
-0.4 & 2.3 & 0.0 & 0.0 & 0.0 & 0.0 & 0.0 & 0.0 & 0.0 & -0.5 & 0.0 & -0.4 & -0.2 & 0.0 & -0.2 & 0.0 & 0.0 & 0.0 & -0.2 \\
-0.2 & 0.0 & 2.0 & -0.2 & 0.0 & 0.0 & -0.5 & 0.0 & 0.0 & -0.2 & 0.0 & 0.0 & 0.0 & -0.2 & -0.2 & 0.0 & 0.0 & 0.0 & -0.2 \\
-0.2 & 0.0 & -0.2 & 2.9 & 0.0 & -0.2 & 0.0 & -0.2 & -0.2 & -0.2 & -0.4 & 0.0 & 0.0 & -0.5 & -0.2 & 0.0 & 0.0 & -0.2 & 0.0 \\
0.0 & 0.0 & 0.0 & 0.0 & 1.2 & 0.0 & 0.0 & 0.0 & 0.0 & 0.0 & 0.0 & -0.2 & 0.0 & 0.0 & -0.2 & -0.2 & -0.2 & -0.2 & 0.0 \\
0.0 & 0.0 & 0.0 & -0.2 & 0.0 & 1.6 & -0.2 & 0.0 & 0.0 & 0.0 & 0.0 & 0.0 & 0.0 & 0.0 & -0.2 & -0.4 & 0.0 & 0.0 & -0.2 \\
0.0 & 0.0 & -0.5 & 0.0 & 0.0 & -0.2 & 2.7 & -0.5 & 0.0 & 0.0 & -0.4 & -0.2 & -0.4 & 0.0 & -0.2 & 0.0 & 0.0 & 0.0 & 0.0 \\
0.0 & 0.0 & 0.0 & -0.2 & 0.0 & 0.0 & -0.5 & 2.7 & -0.2 & -0.2 & 0.0 & 0.0 & 0.0 & 0.0 & 0.0 & -0.4 & -0.4 & 0.0 & -0.2 \\
-0.5 & 0.0 & 0.0 & -0.2 & 0.0 & 0.0 & 0.0 & -0.2 & 1.6 & 0.0 & 0.0 & 0.0 & -0.4 & 0.0 & 0.0 & 0.0 & 0.0 & 0.0 & 0.0 \\
-0.2 & -0.5 & -0.2 & -0.2 & 0.0 & 0.0 & 0.0 & -0.2 & 0.0 & 2.7 & -0.2 & -0.6 & 0.0 & 0.0 & 0.0 & 0.0 & 0.0 & -0.2 & 0.0 \\
-0.2 & 0.0 & 0.0 & -0.4 & 0.0 & 0.0 & -0.4 & 0.0 & 0.0 & -0.2 & 2.8 & -0.4 & 0.0 & -0.2 & -0.7 & 0.0 & 0.0 & 0.0 & 0.0 \\
-0.5 & -0.4 & 0.0 & 0.0 & -0.2 & 0.0 & -0.2 & 0.0 & 0.0 & -0.6 & -0.4 & 3.0 & -0.2 & 0.0 & 0.0 & -0.2 & -0.2 & 0.0 & 0.0 \\
0.0 & -0.2 & 0.0 & 0.0 & 0.0 & 0.0 & -0.4 & 0.0 & -0.4 & 0.0 & 0.0 & -0.2 & 1.9 & -0.2 & -0.5 & 0.0 & 0.0 & 0.0 & 0.0 \\
-0.5 & 0.0 & -0.2 & -0.5 & 0.0 & 0.0 & 0.0 & 0.0 & 0.0 & 0.0 & -0.2 & 0.0 & -0.2 & 1.6 & 0.0 & 0.0 & 0.0 & 0.0 & 0.0 \\
-0.2 & -0.2 & -0.2 & -0.2 & -0.2 & -0.2 & -0.2 & 0.0 & 0.0 & 0.0 & -0.7 & 0.0 & -0.5 & 0.0 & 3.1 & 0.0 & -0.2 & 0.0 & 0.0 \\
-0.2 & 0.0 & 0.0 & 0.0 & -0.2 & -0.4 & 0.0 & -0.4 & 0.0 & 0.0 & 0.0 & -0.2 & 0.0 & 0.0 & 0.0 & 2.2 & -0.2 & -0.2 & 0.0 \\
0.0 & 0.0 & 0.0 & 0.0 & -0.2 & 0.0 & 0.0 & -0.4 & 0.0 & 0.0 & 0.0 & -0.2 & 0.0 & 0.0 & -0.2 & -0.2 & 1.8 & 0.0 & -0.2 \\
0.0 & 0.0 & 0.0 & -0.2 & -0.2 & 0.0 & 0.0 & 0.0 & 0.0 & -0.2 & 0.0 & 0.0 & 0.0 & 0.0 & 0.0 & -0.2 & 0.0 & 1.1 & -0.2 \\
-0.2 & -0.2 & -0.2 & 0.0 & 0.0 & -0.2 & 0.0 & -0.2 & 0.0 & 0.0 & 0.0 & 0.0 & 0.0 & 0.0 & 0.0 & 0.0 & -0.2 & -0.2 & 2.1
\end{pmatrix}
$$

↓  Step 2:(17,13,0)

$$\begin{pmatrix}
3.7 & -0.4 & -0.2 & -0.2 & 0.0 & 0.0 & 0.0 & 0.0 & -0.5 & -0.2 & -0.2 & -0.5 & 0.0 & -0.5 & -0.2 & -0.2 & 0.0 & 0.0 & -0.2 \\
-0.4 & 2.3 & 0.0 & 0.0 & 0.0 & 0.0 & 0.0 & 0.0 & 0.0 & -0.5 & 0.0 & -0.4 & -0.2 & 0.0 & -0.2 & 0.0 & 0.0 & 0.0 & -0.2 \\
-0.2 & 0.0 & 2.0 & -0.2 & 0.0 & 0.0 & -0.5 & 0.0 & 0.0 & -0.2 & 0.0 & 0.0 & 0.0 & -0.2 & -0.2 & 0.0 & 0.0 & 0.0 & -0.2 \\
-0.2 & 0.0 & -0.2 & 2.9 & 0.0 & -0.2 & 0.0 & -0.2 & -0.2 & -0.2 & -0.4 & 0.0 & 0.0 & -0.5 & -0.2 & 0.0 & 0.0 & -0.2 & 0.0 \\
0.0 & 0.0 & 0.0 & 0.0 & 1.2 & 0.0 & 0.0 & 0.0 & 0.0 & 0.0 & 0.0 & -0.2 & 0.0 & 0.0 & -0.2 & -0.2 & -0.2 & -0.2 & 0.0 \\
0.0 & 0.0 & 0.0 & -0.2 & 0.0 & 1.6 & -0.2 & 0.0 & 0.0 & 0.0 & 0.0 & 0.0 & 0.0 & 0.0 & -0.2 & -0.4 & 0.0 & 0.0 & -0.2 \\
0.0 & 0.0 & -0.5 & 0.0 & 0.0 & -0.2 & 2.7 & -0.5 & 0.0 & 0.0 & -0.4 & -0.2 & -0.4 & 0.0 & -0.2 & 0.0 & 0.0 & 0.0 & 0.0 \\
0.0 & 0.0 & 0.0 & -0.2 & 0.0 & 0.0 & -0.5 & 2.7 & -0.2 & -0.2 & 0.0 & 0.0 & 0.0 & 0.0 & 0.0 & -0.4 & -0.4 & 0.0 & -0.2 \\
-0.5 & 0.0 & 0.0 & -0.2 & 0.0 & 0.0 & 0.0 & -0.2 & 1.6 & 0.0 & 0.0 & 0.0 & -0.4 & 0.0 & 0.0 & 0.0 & 0.0 & 0.0 & 0.0 \\
-0.2 & -0.5 & -0.2 & -0.2 & 0.0 & 0.0 & 0.0 & -0.2 & 0.0 & 2.7 & -0.2 & -0.6 & 0.0 & 0.0 & 0.0 & 0.0 & 0.0 & -0.2 & 0.0 \\
-0.2 & 0.0 & 0.0 & -0.4 & 0.0 & 0.0 & -0.4 & 0.0 & 0.0 & -0.2 & 2.7 & -0.4 & 0.0 & -0.2 & -0.7 & 0.0 & 0.0 & 0.0 & 0.0 \\
-0.5 & -0.4 & 0.0 & 0.0 & -0.2 & 0.0 & -0.2 & 0.0 & 0.0 & -0.6 & -0.4 & 3.0 & -0.2 & 0.0 & 0.0 & -0.2 & -0.2 & 0.0 & 0.0 \\
0.0 & -0.2 & 0.0 & 0.0 & 0.0 & 0.0 & -0.4 & 0.0 & -0.4 & 0.0 & 0.0 & -0.2 & 1.9 & -0.2 & -0.5 & 0.0 & 0.0 & 0.0 & 0.0 \\
-0.5 & 0.0 & -0.2 & -0.5 & 0.0 & 0.0 & 0.0 & 0.0 & 0.0 & 0.0 & -0.2 & 0.0 & -0.2 & \textcolor{red}{1.9} & 0.0 & 0.0 & 0.0 & \textcolor{red}{-0.2} & 0.0 \\
-0.2 & -0.2 & -0.2 & -0.2 & -0.2 & -0.2 & -0.2 & 0.0 & 0.0 & 0.0 & -0.7 & 0.0 & -0.5 & 0.0 & 3.1 & 0.0 & -0.2 & 0.0 & 0.0 \\
-0.2 & 0.0 & 0.0 & 0.0 & -0.2 & -0.4 & 0.0 & -0.4 & 0.0 & 0.0 & 0.0 & -0.2 & 0.0 & 0.0 & 0.0 & 2.2 & -0.2 & -0.2 & 0.0 \\
0.0 & 0.0 & 0.0 & 0.0 & -0.2 & 0.0 & 0.0 & -0.4 & 0.0 & 0.0 & 0.0 & -0.2 & 0.0 & 0.0 & -0.2 & -0.2 & 1.8 & 0.0 & -0.2 \\
0.0 & 0.0 & 0.0 & -0.2 & -0.2 & 0.0 & 0.0 & 0.0 & 0.0 & -0.2 & 0.0 & 0.0 & 0.0 & \textcolor{red}{-0.2} & 0.0 & -0.2 & 0.0 & \textcolor{red}{1.4} & -0.2 \\
-0.2 & -0.2 & -0.2 & 0.0 & 0.0 & -0.2 & 0.0 & -0.2 & 0.0 & 0.0 & 0.0 & 0.0 & 0.0 & 0.0 & 0.0 & 0.0 & -0.2 & -0.2 & 2.1
\end{pmatrix}$$

$\downarrow$    Step 3:(8,4,0)

$$\begin{pmatrix}
3.7 & -0.4 & -0.2 & -0.2 & 0.0 & 0.0 & 0.0 & 0.0 & -0.5 & -0.2 & -0.2 & -0.5 & 0.0 & -0.5 & -0.2 & -0.2 & 0.0 & 0.0 & -0.2 \\
-0.4 & 2.3 & 0.0 & 0.0 & 0.0 & 0.0 & 0.0 & 0.0 & 0.0 & -0.5 & 0.0 & -0.4 & -0.2 & 0.0 & -0.2 & 0.0 & 0.0 & 0.0 & -0.2 \\
-0.2 & 0.0 & 2.0 & -0.2 & 0.0 & 0.0 & -0.5 & 0.0 & 0.0 & -0.2 & 0.0 & 0.0 & 0.0 & -0.2 & -0.2 & 0.0 & 0.0 & 0.0 & -0.2 \\
-0.2 & 0.0 & -0.2 & 2.9 & 0.0 & -0.2 & 0.0 & -0.2 & -0.2 & -0.2 & -0.4 & 0.0 & 0.0 & -0.5 & -0.2 & 0.0 & 0.0 & -0.2 & 0.0 \\
0.0 & 0.0 & 0.0 & 0.0 & \textcolor{red}{1.4} & 0.0 & 0.0 & 0.0 & \textcolor{red}{-0.2} & 0.0 & 0.0 & -0.2 & 0.0 & 0.0 & -0.2 & -0.2 & -0.2 & -0.2 & 0.0 \\
0.0 & 0.0 & 0.0 & -0.2 & 0.0 & 1.6 & -0.2 & 0.0 & 0.0 & 0.0 & 0.0 & 0.0 & 0.0 & 0.0 & -0.2 & -0.4 & 0.0 & 0.0 & -0.2 \\
0.0 & 0.0 & -0.5 & 0.0 & 0.0 & -0.2 & 2.7 & -0.5 & 0.0 & 0.0 & -0.4 & -0.2 & -0.4 & 0.0 & -0.2 & 0.0 & 0.0 & 0.0 & 0.0 \\
0.0 & 0.0 & 0.0 & -0.2 & 0.0 & 0.0 & -0.5 & 2.7 & -0.2 & -0.2 & 0.0 & 0.0 & 0.0 & 0.0 & 0.0 & -0.4 & -0.4 & 0.0 & -0.2 \\
-0.5 & 0.0 & 0.0 & -0.2 & \textcolor{red}{-0.2} & 0.0 & 0.0 & -0.2 & \textcolor{red}{1.9} & 0.0 & 0.0 & 0.0 & -0.4 & 0.0 & 0.0 & 0.0 & 0.0 & 0.0 & 0.0 \\
-0.2 & -0.5 & -0.2 & -0.2 & 0.0 & 0.0 & 0.0 & -0.2 & 0.0 & 2.6 & -0.2 & -0.6 & 0.0 & 0.0 & 0.0 & 0.0 & 0.0 & -0.2 & 0.0 \\
-0.2 & 0.0 & 0.0 & -0.4 & 0.0 & 0.0 & -0.4 & 0.0 & 0.0 & -0.2 & 2.7 & -0.4 & 0.0 & -0.2 & -0.7 & 0.0 & 0.0 & 0.0 & 0.0 \\
-0.5 & -0.4 & 0.0 & 0.0 & -0.2 & 0.0 & -0.2 & 0.0 & 0.0 & -0.6 & -0.4 & 3.0 & -0.2 & 0.0 & 0.0 & -0.2 & -0.2 & 0.0 & 0.0 \\
0.0 & -0.2 & 0.0 & 0.0 & 0.0 & 0.0 & -0.4 & 0.0 & -0.4 & 0.0 & 0.0 & -0.2 & 1.9 & -0.2 & -0.5 & 0.0 & 0.0 & 0.0 & 0.0 \\
-0.5 & 0.0 & -0.2 & -0.5 & 0.0 & 0.0 & 0.0 & 0.0 & 0.0 & 0.0 & -0.2 & 0.0 & -0.2 & 1.9 & 0.0 & 0.0 & 0.0 & -0.2 & 0.0 \\
-0.2 & -0.2 & -0.2 & -0.2 & -0.2 & -0.2 & -0.2 & 0.0 & 0.0 & 0.0 & -0.7 & 0.0 & -0.5 & 0.0 & 3.1 & 0.0 & -0.2 & 0.0 & 0.0 \\
-0.2 & 0.0 & 0.0 & 0.0 & -0.2 & -0.4 & 0.0 & -0.4 & 0.0 & 0.0 & 0.0 & -0.2 & 0.0 & 0.0 & 0.0 & 2.2 & -0.2 & -0.2 & 0.0 \\
0.0 & 0.0 & 0.0 & 0.0 & -0.2 & 0.0 & 0.0 & -0.4 & 0.0 & 0.0 & 0.0 & -0.2 & 0.0 & 0.0 & -0.2 & -0.2 & 1.8 & 0.0 & -0.2 \\
0.0 & 0.0 & 0.0 & -0.2 & -0.2 & 0.0 & 0.0 & 0.0 & 0.0 & -0.2 & 0.0 & 0.0 & 0.0 & -0.2 & 0.0 & -0.2 & 0.0 & 1.4 & -0.2 \\
-0.2 & -0.2 & -0.2 & 0.0 & 0.0 & -0.2 & 0.0 & -0.2 & 0.0 & 0.0 & 0.0 & 0.0 & 0.0 & 0.0 & 0.0 & 0.0 & -0.2 & -0.2 & 2.1
\end{pmatrix}$$

$\downarrow$    Step 4:(17,5,1)

$$\begin{pmatrix}
3.7 & -0.4 & -0.2 & -0.2 & 0.0 & 0.0 & 0.0 & 0.0 & -0.5 & -0.2 & -0.2 & -0.5 & 0.0 & -0.5 & -0.2 & -0.2 & 0.0 & 0.0 & -0.2 \\
-0.4 & 2.2 & 0.0 & 0.0 & 0.0 & 0.0 & 0.0 & 0.0 & 0.0 & -0.5 & 0.0 & -0.4 & -0.2 & 0.0 & -0.2 & 0.0 & 0.0 & 0.0 & -0.2 \\
-0.2 & 0.0 & 2.0 & -0.2 & 0.0 & 0.0 & -0.5 & 0.0 & 0.0 & -0.2 & 0.0 & 0.0 & 0.0 & -0.2 & -0.2 & 0.0 & 0.0 & 0.0 & -0.2 \\
-0.2 & 0.0 & -0.2 & 2.8 & 0.0 & -0.2 & 0.0 & -0.2 & -0.2 & -0.2 & -0.4 & 0.0 & 0.0 & -0.5 & -0.2 & 0.0 & 0.0 & -0.2 & 0.0 \\
0.0 & 0.0 & 0.0 & 0.0 & 1.4 & 0.0 & 0.0 & 0.0 & -0.2 & 0.0 & 0.0 & -0.2 & 0.0 & 0.0 & -0.2 & -0.2 & -0.2 & -0.2 & 0.0 \\
0.0 & 0.0 & 0.0 & -0.2 & 0.0 & \textcolor{red}{1.9} & -0.2 & 0.0 & 0.0 & 0.0 & 0.0 & 0.0 & 0.0 & 0.0 & -0.2 & -0.4 & 0.0 & \textcolor{red}{-0.2} & -0.2 \\
0.0 & 0.0 & -0.5 & 0.0 & 0.0 & -0.2 & 2.7 & -0.5 & 0.0 & 0.0 & -0.4 & -0.2 & -0.4 & 0.0 & -0.2 & 0.0 & 0.0 & 0.0 & 0.0 \\
0.0 & 0.0 & 0.0 & -0.2 & 0.0 & 0.0 & -0.5 & 2.7 & -0.2 & -0.2 & 0.0 & 0.0 & 0.0 & 0.0 & 0.0 & -0.4 & -0.4 & 0.0 & -0.2 \\
-0.5 & 0.0 & 0.0 & -0.2 & -0.2 & 0.0 & 0.0 & -0.2 & 1.9 & 0.0 & 0.0 & 0.0 & -0.4 & 0.0 & 0.0 & 0.0 & 0.0 & 0.0 & 0.0 \\
-0.2 & -0.5 & -0.2 & -0.2 & 0.0 & 0.0 & 0.0 & -0.2 & 0.0 & 2.6 & -0.2 & -0.6 & 0.0 & 0.0 & 0.0 & 0.0 & 0.0 & -0.2 & 0.0 \\
-0.2 & 0.0 & 0.0 & -0.4 & 0.0 & 0.0 & -0.4 & 0.0 & 0.0 & -0.2 & 2.7 & -0.4 & 0.0 & -0.2 & -0.7 & 0.0 & 0.0 & 0.0 & 0.0 \\
-0.5 & -0.4 & 0.0 & 0.0 & -0.2 & 0.0 & -0.2 & 0.0 & 0.0 & -0.6 & -0.4 & 3.0 & -0.2 & 0.0 & 0.0 & -0.2 & -0.2 & 0.0 & 0.0 \\
0.0 & -0.2 & 0.0 & 0.0 & 0.0 & 0.0 & -0.4 & 0.0 & -0.4 & 0.0 & 0.0 & -0.2 & 1.9 & -0.2 & -0.5 & 0.0 & 0.0 & 0.0 & 0.0 \\
-0.5 & 0.0 & -0.2 & -0.5 & 0.0 & 0.0 & 0.0 & 0.0 & 0.0 & 0.0 & -0.2 & 0.0 & -0.2 & 1.9 & 0.0 & 0.0 & 0.0 & -0.2 & 0.0 \\
-0.2 & -0.2 & -0.2 & -0.2 & -0.2 & -0.2 & -0.2 & 0.0 & 0.0 & 0.0 & -0.7 & 0.0 & -0.5 & 0.0 & 3.1 & 0.0 & -0.2 & 0.0 & 0.0 \\
-0.2 & 0.0 & 0.0 & 0.0 & -0.2 & -0.4 & 0.0 & -0.4 & 0.0 & 0.0 & 0.0 & -0.2 & 0.0 & 0.0 & 0.0 & 2.2 & -0.2 & -0.2 & 0.0 \\
0.0 & 0.0 & 0.0 & 0.0 & -0.2 & 0.0 & 0.0 & -0.4 & 0.0 & 0.0 & 0.0 & -0.2 & 0.0 & 0.0 & -0.2 & -0.2 & 1.8 & 0.0 & -0.2 \\
0.0 & 0.0 & 0.0 & -0.2 & -0.2 & \textcolor{red}{-0.2} & 0.0 & 0.0 & 0.0 & -0.2 & 0.0 & 0.0 & 0.0 & -0.2 & 0.0 & -0.2 & 0.0 & \textcolor{red}{1.6} & -0.2 \\
-0.2 & -0.2 & -0.2 & 0.0 & 0.0 & -0.2 & 0.0 & -0.2 & 0.0 & 0.0 & 0.0 & 0.0 & 0.0 & 0.0 & 0.0 & 0.0 & -0.2 & -0.2 & 2.1
\end{pmatrix}$$

$\downarrow$    Step 5:(4,2,0.5)

$$\begin{pmatrix}
3.7 & -0.4 & -0.2 & -0.2 & 0.0 & 0.0 & 0.0 & 0.0 & -0.5 & -0.2 & -0.2 & -0.5 & 0.0 & -0.5 & -0.2 & -0.2 & 0.0 & 0.0 & -0.2 \\
-0.4 & 2.2 & 0.0 & 0.0 & 0.0 & 0.0 & 0.0 & 0.0 & 0.0 & -0.5 & 0.0 & -0.4 & -0.2 & 0.0 & -0.2 & 0.0 & 0.0 & 0.0 & -0.2 \\
-0.2 & 0.0 & \textcolor{red}{2.2} & -0.2 & \textcolor{red}{-0.2} & 0.0 & -0.5 & 0.0 & 0.0 & -0.2 & 0.0 & 0.0 & 0.0 & -0.2 & -0.2 & 0.0 & 0.0 & 0.0 & -0.2 \\
-0.2 & 0.0 & -0.2 & 2.8 & 0.0 & -0.2 & 0.0 & -0.2 & -0.2 & -0.2 & -0.4 & 0.0 & 0.0 & -0.5 & -0.2 & 0.0 & 0.0 & -0.2 & 0.0 \\
0.0 & 0.0 & \textcolor{red}{-0.2} & 0.0 & \textcolor{red}{1.7} & 0.0 & 0.0 & 0.0 & -0.2 & 0.0 & 0.0 & -0.2 & 0.0 & 0.0 & -0.2 & -0.2 & -0.2 & -0.2 & 0.0 \\
0.0 & 0.0 & 0.0 & -0.2 & 0.0 & 1.8 & -0.2 & 0.0 & 0.0 & 0.0 & 0.0 & 0.0 & 0.0 & 0.0 & -0.2 & -0.4 & 0.0 & -0.2 & -0.2 \\
0.0 & 0.0 & -0.5 & 0.0 & 0.0 & -0.2 & 2.6 & -0.5 & 0.0 & 0.0 & -0.4 & -0.2 & -0.4 & 0.0 & -0.2 & 0.0 & 0.0 & 0.0 & 0.0 \\
0.0 & 0.0 & 0.0 & -0.2 & 0.0 & 0.0 & -0.5 & 2.7 & -0.2 & -0.2 & 0.0 & 0.0 & 0.0 & 0.0 & 0.0 & -0.4 & -0.4 & 0.0 & -0.2 \\
-0.5 & 0.0 & 0.0 & -0.2 & -0.2 & 0.0 & 0.0 & -0.2 & 1.8 & 0.0 & 0.0 & 0.0 & -0.4 & 0.0 & 0.0 & 0.0 & 0.0 & 0.0 & 0.0 \\
-0.2 & -0.5 & -0.2 & -0.2 & 0.0 & 0.0 & 0.0 & -0.2 & 0.0 & 2.6 & -0.2 & -0.6 & 0.0 & 0.0 & 0.0 & 0.0 & 0.0 & -0.2 & 0.0 \\
-0.2 & 0.0 & 0.0 & -0.4 & 0.0 & 0.0 & -0.4 & 0.0 & 0.0 & -0.2 & 2.6 & -0.4 & 0.0 & -0.2 & -0.7 & 0.0 & 0.0 & 0.0 & 0.0 \\
-0.5 & -0.4 & 0.0 & 0.0 & -0.2 & 0.0 & -0.2 & 0.0 & 0.0 & -0.6 & -0.4 & 2.9 & -0.2 & 0.0 & 0.0 & -0.2 & -0.2 & 0.0 & 0.0 \\
0.0 & -0.2 & 0.0 & 0.0 & 0.0 & 0.0 & -0.4 & 0.0 & -0.4 & 0.0 & 0.0 & -0.2 & 1.8 & -0.2 & -0.5 & 0.0 & 0.0 & 0.0 & 0.0 \\
-0.5 & 0.0 & -0.2 & -0.5 & 0.0 & 0.0 & 0.0 & 0.0 & 0.0 & 0.0 & -0.2 & 0.0 & -0.2 & 1.9 & 0.0 & 0.0 & 0.0 & -0.2 & 0.0 \\
-0.2 & -0.2 & -0.2 & -0.2 & -0.2 & -0.2 & -0.2 & 0.0 & 0.0 & 0.0 & -0.7 & 0.0 & -0.5 & 0.0 & 3.0 & 0.0 & -0.2 & 0.0 & 0.0 \\
-0.2 & 0.0 & 0.0 & 0.0 & -0.2 & -0.4 & 0.0 & -0.4 & 0.0 & 0.0 & 0.0 & -0.2 & 0.0 & 0.0 & 0.0 & 2.2 & -0.2 & -0.2 & 0.0 \\
0.0 & 0.0 & 0.0 & 0.0 & -0.2 & 0.0 & 0.0 & -0.4 & 0.0 & 0.0 & 0.0 & -0.2 & 0.0 & 0.0 & -0.2 & -0.2 & 1.8 & 0.0 & -0.2 \\
0.0 & 0.0 & 0.0 & -0.2 & -0.2 & -0.2 & 0.0 & 0.0 & 0.0 & -0.2 & 0.0 & 0.0 & 0.0 & -0.2 & 0.0 & -0.2 & 0.0 & 1.6 & -0.2 \\
-0.2 & -0.2 & -0.2 & 0.0 & 0.0 & -0.2 & 0.0 & -0.2 & 0.0 & 0.0 & 0.0 & 0.0 & 0.0 & 0.0 & 0.0 & 0.0 & -0.2 & -0.2 & 2.1
\end{pmatrix}$$

