# OpenReview forum: "Fewer Battles, More Gain: An Information-Efficient Framework for Arena-based LLM Evaluation"
_ICLR.cc/2026/Conference — ICLR 2026 Poster_

### Official Review · Reviewer_d3Aw · 2025-10-30

**Soundness:** 3
**Presentation:** 3
**Contribution:** 3
**Rating:** 6
**Confidence:** 3

**Summary:**

The paper introduces an information-theoretic approach to Arena-based LLM evaluation, applying Fisher information and optimal experimental design (A-/D-optimality) to select the most informative matchups.

**Strengths:**

1. This paper is a elegant extension of optimal design principles to LLM ranking tasks. The inclusion of Top-K concurrent battles and adaptive scheduling shows good awareness of practical constraints in large-scale evaluation pipelines.
2. The experiments demonstrate that the information-efficient pairing strategy achieves faster convergence and higher ranking consistency with fewer matches. The information-based metrics (A-info and D-info) are intuitive and interpretable for monitoring evaluation efficiency.

**Weaknesses:**

1. The “ground truth” ranking is derived from the same model used in the method, introducing potential circularity. It would strengthen the paper to include an external or human-verified ranking baseline to validate real-world correlation.
2. The asymptotic normality and information growth results assume an Erdős–Rényi random graph of matchups. However, the real algorithm chooses pairs adaptively based on prior results, violating the independence assumption.

**Questions:**

1. The explanation for Figure 5 (Line 468 - Line 473) is confusing. Can you explain more details?

---

> ### Author Response · Authors · 2025-11-18
>
> Thank you for your valuable feedback! Regarding the questions you raised, we have carefully considered each point and have made the following responses:
>
> > **Q1**: The “ground truth” ranking is derived from the same model used in the method, introducing potential circularity. It would strengthen the paper to include an external or human-verified ranking baseline to validate real-world correlation.
>
> We acknowledge that the lack of an external or human-verified ground truth is a limitation of our work. The Arena-based Evaluation itself is designed to produce model rankings that reflect human preferences through pairwise comparisons. Obtaining an additional human-verified ranking would introduce another group of subjective preferences. Practically, as described in lines 306–316 of our paper, our ground-truth ranking can be interpreted as follows: due to the asymptotic normality guaranteed even in sparse settings, our dataset can be seen as approximating a scenario with a very large |S|. In this regime, the estimated ranking (our ground truth) converges to the true ranking. Our goal is to achieve an estimated ranking close to this “true” ranking using a small number of sampled matches. Such a ground-truth approach is also adopted in other data-efficiency or ability-evaluation studies [1]. To avoid ambiguity, we have provided a more detailed description of the evaluation metrics at line 316: "In other words, our task can be viewed... improving efficiency"
>
> > **Q2**: The asymptotic normality and information growth results assume an Erdős–Rényi random graph of matchups. However, the real algorithm chooses pairs adaptively based on prior results, violating the independence assumption.
>
> Thank you for pointing this out. We acknowledge that there is a gap between theory and practice.
>
> However, our theoretical results are not meant to model the exact sampling process of the adaptive algorithm, but rather to provide a principled way to define Fisher information and to understand how information should accumulate when comparisons are sufficiently diverse.
>
> Empirically, we observe that (i) the Fisher information under the adaptive procedure still grows almost linearly, and (ii) the estimated rankings converge stably despite the dependence in pair selection. These behaviors suggest that the theoretical framework remains a meaningful approximation and offers valuable intuition, even if the independence assumption is not strictly satisfied.
>
> Moreover, similar situations, where iterative selection conflicts with independence assumptions, also arise in many adaptive domains, such as active learning and computerized adaptive testing, with comparable approaches for handling them [1] [2]. This suggests that, even though the independence assumptions are not strictly satisfied, the adaptive procedures can still provide meaningful and informative results.
>
> > **Q3**: The explanation for Figure 5 (Line 468 - Line 473) is confusing. Can you explain more details?
>
> We apologize for any confusion. In fact, Figure 5 illustrates the time cost of our algorithm. Through the two improvements described in lines 284–290, we have controlled the time cost to an acceptable level—for instance, it can select suitable model pair within 1 second from 200 models. Our explanation in the paper aims to clarify why the current cost is acceptable: the evaluation process involves not only selection, but also user queries, LLM generation, and user annotations. Within this process, the time spent on selection is negligible. To avoid potential confusion, we have updated Lines 468–473 in the revised version of the paper.
>
> [1] A Bounded Ability Estimation for Computerized Adaptive Testing. 2023
>
> [2] Characterizing the robustness of Bayesian adaptive experimental designs to active learning bias, 2022

---

### Official Review · Reviewer_Mho6 · 2025-10-30

**Soundness:** 4
**Presentation:** 4
**Contribution:** 3
**Rating:** 8
**Confidence:** 3

**Summary:**

This paper introduces a method to improve the efficiency of model comparisons in a chatbot arena style setting consisting of repeated pairwise comparisons between models. Specifically, it uses the Fisher information of a matrix containing information about previous comparisons to guide selection of model pairs to consider. They introduce two specific metrics using Fisher Information, and show on real-world data that using them for model pair selection improves efficiency while still being robust and reliable.

**Strengths:**

1. I think the problem of improving the efficiency and reliability of pairwise evaluations is incredibly compelling yet under-explored. There is a good amount of work exploring the efficiency problem when considering large benchmarks, but there are fewer that look at the "arena" style comparisons, which are nonetheless very popular.
2. The experiments are thorough and consider real-world data at a reasonable scale. Multiple baseline methods are compared against, as well as multiple variants of ELO.
3. The paper contains a rigorous theoretical argument for their approach.

**Weaknesses:**

1. While I think the baselines are well-chosen, I would have liked to have seen some more comparisons with other active learning methods. I'd especially be interested in comparisons regarding how much data is required for initialization for other methods, and if this new method addresses the cold start problem in a much more efficient way.
2. The relative performance gains don't seem too substantial considering the additional cost of the method. I think the authors address the caveats in their comparisons well, but I think some further analysis would be warranted to get a finer-grained understanding of the trade-offs between additional computational resources required at scale for the method and annotation volume.
3. I think one major assumption made here is a fixed prompt distribution. Intuitively, I'd expect the specific test instance chosen to have a major impact as well on determining the relative quality of models. I understand this is likely beyond the scope of this paper, but I think some comparisons with methods that perhaps only consider this factor would be useful, since it could then help compare the impact of properties of the test input and properties of the model pairs themselves.

**Questions:**

1. In the visualization section, you mention that D-Optimality might just be better in general in the arena setting than A-optimality. Is there an intuition behind what scenarios A-optimality might be better?
2. How much impact does using a smaller initialization set have?

---

> ### Author Response · Authors · 2025-11-18
>
> Thank you for your thoughtful feedback! We have carefully examined your questions and provide our detailed responses below:
>
> > **Q1**: I would have liked to have seen some more comparisons with other active learning methods.
>
> While our task shares similarities with active learning, its unique characteristics (such as the ability to select the same model pair multiple times to obtain distinct annotation results, and the relative nature of model parameters) prevent direct application of cutting-edge methods like gradient matching[1] in our scenario. Nevertheless, the strategy of leveraging expected model change(EMC)[2] is applicable here, and we have thus supplemented the following experiments:
>
> | Method  | Chatbot    | PPE        | Real       | Simu       | ELO        | m-ELO      | Total      |
> | ------- | ---------- | ---------- | ---------- | ---------- | ---------- | ---------- | ---------- |
> | Nearest | 0.8825     | 0.7995     | 0.8422     | 0.8398     | 0.8412     | 0.8407     | 0.8410     |
> | EMC     | 0.8810     | 0.8021     | 0.8446     | 0.8385     | 0.8412     | 0.8418     | 0.8415     |
> | A-optim | 0.8880     | 0.8033     | 0.8494     | 0.8418     | 0.8406     | 0.8506     | 0.8456     |
> | D-optim | **0.8941** | **0.8122** | **0.8596** | **0.8466** | **0.8486** | **0.8577** | **0.8531** |
>
> Notably, the expected model change (EMC) method is only suitable for m-ELO. In traditional ELO, model changes can be explicitly calculated, which reduces EMC to the Nearest method. As we observed, this method performs similarly to the Nearest method, but still falls noticeably short of the method we propose. Moreover, although EMC requires only a comparable amount of initial data as D-optim, it measures distances in the latent space rather than the probability space and therefore relies heavily on the accuracy of the current ability estimates. As a result, EMC is not well suited for cold-start scenarios.
>
> > **Q2**: some further analysis would be warranted to get a finer-grained understanding of the trade-offs between additional computational resources required at scale for the method and annotation volume.
>
> Actually, our main purpose is to trade computational resources for human effort, which is particularly practical since the resources required by our method are much smaller compared to the inference cost of large models. In the revised version of the paper, we have updated Lines 468–473 to better clarify our discussion of the time cost associated with sample selection.
>
> > **Q3**: I think one major assumption made here is a fixed prompt distribution.
>
> This is also something we hope to explore in the future. In our view, if test instances are to be incorporated, the static evaluation framework must first be revised to account for instance-level factors within our modeling process (m-ELO). For example, one could integrate the discriminative power of each test instance into the ELO update. If this can be achieved, our algorithm could, in principle, jointly select instances, models, and even annotators, enabling a fully adaptive evaluation paradigm. We have included this part of the discussion as future work in the **Conclusion** section: "In future work, an important direction is ... test instances, models, and even annotators."
>
> > **Q4**: In the visualization section, you mention that D-Optimality might just be better in general in the arena setting than A-optimality. Is there an intuition behind what scenarios A-optimality might be better?
>
> Yes, the comparison between A-optimality and D-optimality has been extensively studied in the design of experiments, and many previous works have examined these two approaches and discuss their advantages [3] [4] [5]. For this reason, our paper does not discard either approach; instead, we provide discussion and analysis of both.
>
> > **Q5**: How much impact does using a smaller initialization set have?
>
> If the initial set only satisfies the requirement in Lemma 1(1), it may produce a stable estimated ranking, but this does not guarantee that the estimated ranking closely matches the true ranking. If the sample size is too close to the bound, abnormal situations may occur. For an extreme example, if a model performs poorly during initialization or if anomalous annotation behavior occurs, resulting in an unusually low estimated ability, then selecting models based on this ability may be counterproductive, requiring more time to adjust the model to its correct position.
>
> Once again, we sincerely thank you for your valuable questions and suggestions. If you have any further inquiries, please feel free to reach out.
>
> [1] GRAD-MATCH: Gradient Matching based Data Subset Selection for Efficient Deep Model Training, 2021s
>
> [2] Maximizing expected model change for active learning in regression, 2013
>
> [3] A-optimal versus D-optimal design of screening experiments, 2021
>
> [4] D- and A-Optimal Screening Designs, 2023
>
> [5] Comparing robust properties of A, D, E and G-optimal designs,1994

---

> > ### Comment · Reviewer_Mho6 · 2025-11-19
> >
> > Thank you for the helpful clarifications and new experiments. Regarding Q3, there has been some work on incorporating discriminative power of prompts into ELO scores https://aclanthology.org/2024.findings-emnlp.747/ which you might find interesting.

---

> > > ### Author Response · Authors · 2025-11-24
> > >
> > > Thank you again for sharing your insights on the *discriminative power of prompts*, which have been highly valuable for guiding our subsequent research. In our current study, we attempt to incorporate item-specific parameters and conduct a series of experiments under this setting. Specifically, we assume that the test instances are fixed prior to model selection, and we select appropriate model pairs based on both the given test instance and historical evaluation records. For the sake of convenience, we directly sample from a log-normal distribution to determine the discriminative parameter of each test instance, denoted as $\alpha_t$.
> > >
> > > We change the ELO probability density function from
> > >
> > > $$p_{ij}=\frac{1}{1+e^{-C(r_i-r_j)}}$$ to $$p_{ij}=\frac{1}{1+e^{-\alpha_t(r_i-r_j)}}$$
> > >
> > > This design is consistent with the idea in the paper you mentioned, namely that prompts with higher discriminative power correspond to a larger effective learning rate. Under this formulation, the information gain in our method is also adjusted from
> > >
> > > $\sum C^2 p_{ij} p_{ji} e_i e_j^\top$ to $\sum \alpha_t^2 p_{ij} p_{ji} e_i e_j^\top$
> > >
> > > We conduct experiments on the Chatbot dataset, and the experimental results are presented as follows:
> > >
> > > | Method   | T=100      | T=200      | T=500      | T=1000     |
> > > | -------- | ---------- | ---------- | ---------- | ---------- |
> > > | Random   | 0.8779     | 0.9105     | 0.9284     | 0.9516     |
> > > | Nearest  | 0.8695     | 0.8842     | 0.8884     | 0.8916     |
> > > | Ada-Pair | 0.8716     | 0.8989     | 0.9463     | 0.9632     |
> > > | A-optim  | **0.8958** | 0.9137     | 0.9495     | 0.9653     |
> > > | D-optim  | 0.8884     | **0.9189** | **0.9484** | **0.9758** |
> > >
> > > As can be observed, the improved Fisher information–based method achieves a significant overall performance gain, indicating that our approach, after incorporating the modified m-ELO framework, is capable of effectively accounting for the discriminative power of test instances.
> > >
> > > In future work, we plan to further extend this line of research. In particular, we aim to address a current limitation of our method, which can only map from *text to discriminative power* but lacks the inverse capability of *generating text from discriminative power*. Our goal is to develop a unified framework that can simultaneously select model pairs and dynamically generate appropriate test instances.
> > >
> > > Once again, we sincerely thank you for your valuable suggestions and guidance. Your insights have been highly meaningful and instrumental to our work.

---

### Official Review · Reviewer_xuUo · 2025-11-01

**Soundness:** 4
**Presentation:** 4
**Contribution:** 4
**Rating:** 8
**Confidence:** 4

**Summary:**

This paper addresses the inefficiency of current arena-based evaluations for Large Language Models (LLMs), such as Chatbot Arena. At present, these arenas depend on random or exhaustive model pairings for human preference annotation, which results in significant redundant evaluations and a waste of resources.

To address this, the paper proposes an adaptive model-pair selection algorithm. The core of this method leverages the statistical properties of ELO rating estimation (asymptotic normality) and introduces Fisher Information as a metric to actively select the most informative (i.e., highest "value") model pairs. Specifically, the paper proposes: A-optimality: Aims to minimize the total variance of the model ability estimations. And D-optimality: Aims to maximize the determinant of the Fisher Information Matrix (FIM) to globally reduce the uncertainty in model rankings.

**Strengths:**

The problem formulation is significant and compelling.

The method is validated effectively through experiments.

**Weaknesses:**

The paper's theory is entirely established upon the m-ELO framework, with the FIM being derived from its likelihood function. However, in the experiments, the A/D-optimality strategies, which are based on this specific FIM, are applied to both m-ELO and traditional ELO. This is problematic because traditional ELO is an iterative update algorithm and does not directly optimize a global log-likelihood function.

The robustness of the heuristic experiments needs further validation.

**Questions:**

A-optimality demonstrates a severe sampling bias in the experiments, showing an extreme tendency to select pairs involving the "anchor" model. Why does D-optimality manage to avoid this issue?

---

> ### Author Response · Authors · 2025-11-18
>
> We thank the reviewer for the insightful and constructive comments. We address the two main concerns as follows:
>
> > **Q1:** The paper's theory is entirely established upon the m-ELO framework, with the FIM being derived from its likelihood function. However, in the experiments, the A/D-optimality strategies, which are based on this specific FIM, are applied to both m-ELO and traditional ELO. This is problematic because traditional ELO is an iterative update algorithm and does not directly optimize a global log-likelihood function.
>
> We fully agree that our theoretical derivation of the Fisher Information Matrix (FIM) is based on the maximum-likelihood formulation in m-ELO. However, the traditional ELO algorithm can be viewed as an online approximation of this process: each ELO update corresponds to a stochastic gradient step on a single-sample log-likelihood. Therefore, applying the FIM-based selection criteria to ELO remains conceptually consistent—each iteration implicitly estimates player (model) skills under the same probabilistic assumptions.
>
> Therefore, we attempted to apply our proposed adaptive selection method to the traditional ELO estimation. We found that it actually improves ranking accuracy in this setting as well, demonstrating the method’s general applicability across different estimation frameworks.
>
> > **Q2**: The robustness of the heuristic experiments needs further validation.
>
> Thank you for your reminder! How to design an effective and robust experimental scheme for this research is a question that we urgently need to explore in depth. Considering that each experiment underwent multiple rounds of verification, the robustness of our results is well ensured. In addition, we have supplemented our study with experiments on the sampling methods used in the simulation, as shown in the table below.
>
> | Step=1000,Dataset=Chatbot | C/C*=0.5   | 1          | 2          | inf        |
> | ------------------------- | ---------- | ---------- | ---------- | ---------- |
> | Random                    | 0.8537     | 0.9095     | 0.9526     | 1.0000     |
> | Nearest                   | 0.8253     | 0.9179     | 0.9242     | 1.0000     |
> | Ada-Pair                  | 0.8453     | 0.9242     | 0.9621     | 1.0000     |
> | A-optim                   | 0.8295     | 0.9263     | 0.9516     | 1.0000     |
> | D-optim                   | **0.8537** | **0.9284** | **0.9579** | **1.0000** |
>
> In the table, the hyperparameter C/C^* denotes the ratio relative to the value used in the original experiment where C* is the parameter for ELO; a larger ratio indicates a lower probability of outcome reversal. “inf” means that the stronger model always wins. As the results show, regardless of whether the result will flip or not, D-optimal consistently delivers superior performance, further demonstrating the robustness of the method.
>
> > **Q3**: A-optimality demonstrates a severe sampling bias in the experiments, showing an extreme tendency to select pairs involving the "anchor" model. Why does D-optimality manage to avoid this issue?
>
> We acknowledge that A-optimality tends to exhibit sampling bias, frequently involving “anchor” models in selected pairs. Numerically, this occurs because A-optimality can rapidly reduce its objective by focusing on directions with the largest variance.In contrast, D-optimality maximizes the determinant of the FIM, which geometrically minimizes the volume of the confidence ellipsoid—effectively achieving isotropic shrinkage in all directions. Consequently, D-optimality promotes more balanced information gain and more uniform pair selection, as also reflected in our experiments [1] [2].
>
> Meanwhile, we can understand this phenomenon based on the following example：suppose we have three models A, B, and C with identical abilities, and the ability of model C is fixed—resulting in a 2×2 information matrix. Let the pairwise battle frequencies between (A,B), (B,C), and (C,A) be a, b, and c respectively (where \(a + b + c = 1\)). It can be observed that the D-optimal criterion attains its maximum at \(1/3, 1/3, 1/3), while the A-optimal criterion does not yield its extremum at this point. That means, even with model C fixed, D-optimal still maintains uniform sampling, whereas A-optimal fails to do so. This indicates that A-optimal cannot guarantee the property of uniformity even the number of models is small.
>
> We hope these responses address your concerns. If you have any further questions, please feel free to let us know.
>
> [1] A-optimal versus D-optimal design of screening experiments, 2021
>
> [2] D- and A-Optimal Screening Designs, 2023

---

### Meta-Review · Area_Chair_FqZS · 2026-01-01

**Summary:**

Strengths mentioned by the reviewers:
- Problem formulation is compelling
- Method is validated effectively during the experiments.

Weaknesses mentioned by the reviewers:
- The method is established on the m-ELO framework. However, A/D-optimality strategies are applied to both, m-ELO and ELO. **Addressed.**
- How do you compare to with other active learning methods? How much data is required for initialization for other methods? Can they avoid the cold start problem? **Addressed**
- The relative performance gain does not seem too substantial considering the additional cost of the method. **Not addressed.**

Questions:
- A-optimality demonstrates a severe sampling bias in the experiments, showing an extreme tendency to select pairs involving the "anchor" model. Why does D-optimality manage to avoid this issue? **Addressed**
- Is there an intuition behind what scenarios A-optimality might be better? **Addressed**
- How much impact does using a smaller initialization set have? **Partially addressed**
- The “ground truth” ranking is derived from the same model used in the method, introducing potential circularity. It would strengthen the paper to include an external or human-verified ranking baseline to validate real-world correlation. **Addressed**
- The asymptotic normality and information growth results assume an Erdős–Rényi random graph of matchups. However, the real algorithm chooses pairs adaptively based on prior results, violating the independence assumption. **Partially addressed.** Authors acknowledge this gap between theory and practice.
- The explanation for Figure 5 (Line 468 - Line 473) is confusing. Can you explain more details? **Partially addressed.**

**Reviewer Concerns:**

While may concerns the reviewers raised where fully or at least partially addressed, one remained unaddressed.

One reviewer raised the concern that the performance gains appear to not be substantial. Specifically Figure 2: While `D-optim` has sometimes the highest score, the error bars (declared in the caption to be the standard deviation) do not support the statement that the improvement this is either consistent nor significant at a useful confidence level (i.e. $\geq 0.95$). In Table 2, the pairwise metric scores are reported without confidence intervals, which would be crucial in order to claim statistical significance. The ideas of the paper are interesting and worth sharing.

**Reviewer Scores:**

The reviewers would have retained their ratings as most concerns where addressed.

---

### Decision · Program_Chairs · 2026-01-26

Accept (Poster)